# HalluText: Towards Benchmarking and Mitigating OCR Hallucination for LVLMs

## Abstract

Optical Character Recognition (OCR) serves as a critical bridge connecting vision and language, attracting increasing attention in the community of Large Vision-Language Models (LVLMs). However, due to the prevalent encode-then-decode architecture, LVLMs tend to over-rely on language priors, leading to frequent failures in following basic visual-text instructions. We term this issue OCR hallucination. To systematically mitigate it and facilitate reliable OCR perception in LVLMs, we conduct the first large-scale empirical analysis based on OCRBench v2. Our findings reveal that current LVLMs frequently misinterpret or ignore textual visual content, particularly across two orthogonal dimensions, including perception task and hallucination taxonomy. Building on these insights, we introduce HalluText, a benchmark specifically designed to comprehensively evaluate OCR hallucination in LVLMs across nine subclasses. Alongside this benchmark, we propose OCRAssistor, a lightweight plug-and-play method pioneering large-small model collaboration. By integrating compact OCR model outputs into the LVLM decoding process, it achieves a 9.6% improvement on HalluText with only marginal computational cost. When applied to OCRBench v2, this method also improves the performance of the top-performing open-source model Qwen2.5-VL-7B, achieving a 3% gain and highlighting the importance of addressing OCR hallucination in LVLMs. Through our benchmark and proposed solution, we hope to shed light on the challenges and potential pathways for improving visual text perception in LVLMs. The organized benchmark and the relevant code will be released soon.

## 1 Introduction

Driven by advances from both academia and industry, Large Vision Language Models (LVLMs) are increasingly applied across a wide range of domains. As a crucial bridge between vision and language, Optical Character Recognition (OCR) has emerged as both a foundational pre-training paradigm and a key task for supervised fine-tuning. OCR-centric tasks have also garnered significant attention from both general-purpose (Wang et al., 2024b; Li et al., 2024a; Lu et al., 2024; Yao et al., 2024; Bai et al., 2025; Zhu et al., 2025a) and OCR-specialized LVLMs (Li et al., 2024b; Huang et al., 2024a; Yu et al., 2024b; Zhao et al., 2024; Nacson et al., 2025; Li et al., 2025), owing to their wide applicability in real-world scenarios such as smart offices, content moderation, and document intelligence.

Despite this growing focus on OCR-centric tasks, we observe that current LVLMs still often struggle with seemingly simple questions that involve understanding text within images. Figure 1 illustrates three representative failure cases where state-of-the-art models (Yao et al., 2024; Huang et al., 2024a; Bai et al., 2025) consistently fail. Borrowing the concept of hallucinations, we attribute this issue as "**OCR hallucination**", defined as instances where the responses generated by LVLMs fail to accurately follow visual text-centered instructions. To systematically analyze and attribute these errors, we conduct a comprehensive empirical study on widely adopted OCRBench v2 (Fu et al., 2024). Evaluations across over 1,000 samples reveal that these errors are prevalent across different LVLMs and tasks, while exhibiting consistent patterns that enable their categorical grouping. Driven by such findings, we categorize the errors along two orthogonal dimensions: 1) **the perception task stage**, which focuses on the perceptual stages of localization and recognition, and 2) **the hallucination taxonomy**, which classifies error types into category, relation, and attribute hallucinations. Take the

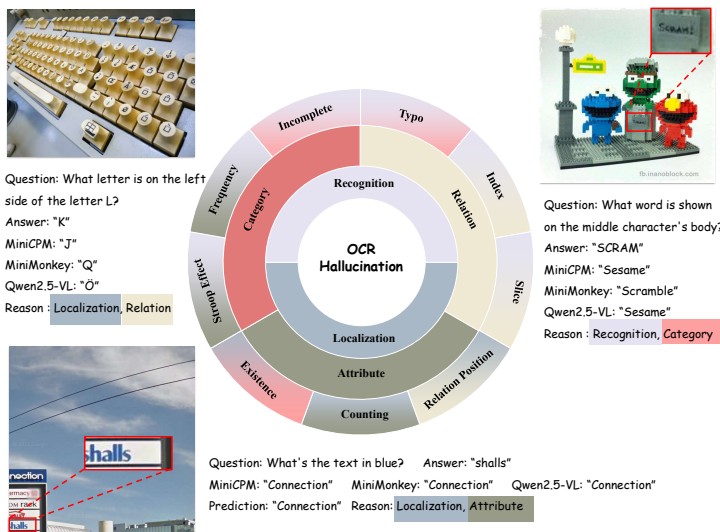

Figure 1: Taxonomy of OCR Hallucinations. The inner rings represent a dual-perspective taxonomy by task stage and hallucination type, while the outer ring indicates the nine HalluText subsets aligned with these categories. Colors denote hallucination types, and three examples around the perimeter illustrate their definitions and characteristics.

the bottom of Figure 1 as an example, the question is *"What's the text in blue?"*, the correct answer is *"shalls"*, but all three models erroneously output *"Connection"*. We attribute this error primarily to incorrect localization—the models fail to attend to the region containing the blue text. Further analysis reveals that this localization failure stems from a misunderstanding of the visual attribute "blue", indicating an error due to attribute hallucination. Thus, this case exemplifies how an initial perceptual failure (e.g., color misinterpretation) can propagate to semantic-level hallucinations. By framing OCR errors within this dual-perspective framework, we aim to better understand the underlying causes of OCR failures and provide actionable insights for future model development.

Based on these insights, we introduce a new benchmark **HalluText**. Unlike the scattered hallucination-related samples in OCRBench v2, HalluText offers a more comprehensive and structured diagnosis for OCR hallucination in LVLMs. Following (Yin et al., 2024), we formulate test samples as multiple-choice questions and collect 4,678 image–question–answer triplets covering nine distinct types of hallucinations. As illustrated in Figure 1, these types are built upon the two previous orthogonal dimensions, comprising existence, incompletion, typo, position, index, slice, counting, frequency, and stroop effect.

Furthermore, we also propose a lightweight, plug-and-play method to mitigate OCR hallucinations, called **OCRAssistor**. This framework pioneers a novel collaborative paradigm between large and small models, where a small-scale OCR-specialized model injects vision-grounded cues to guide the decoding process of large vision-language models (LVLMs). Notably, this design does not require additional fine-tuning of the large model, making it both efficient and flexible to deploy. Despite its simplicity, OCRAssistor achieves impressive results, improving the baseline Qwen2.5-VL-7B by 9.6% on HalluText. When applied to the more general OCRBench v2, it outperforms the baseline by 2.5% on the English subset and 3.7% on the Chinese subset. These results not only demonstrate the effectiveness and generalizability of our approach in fine-grained perception tasks but also underscore the critical importance of addressing hallucination in OCR-centric applications. Our contributions are summarized into three main aspects.

- We conduct an extensive empirical study to uncover the overlooked problem of **OCR hallucination** in LVLMs. Driven by the results, we establish a dual-perspective taxonomy based on the task categories and hallucination types to systematically analyze these errors.

- We construct **HalluText**, a fine-grained benchmark for OCR hallucination. HalluText consists of 4,678 carefully curated samples across 9 subsets, each targeting specific perception and hallucina-

tion dimensions. Compared to existing benchmarks like OCRBench v2, HalluText offers a more comprehensive and structured diagnostic of OCR hallucination, providing clear insights for future advancements of LVLMs.

- We design **OCRAssistor**, a plug-and-play method to mitigate OCR hallucination through a novel large-small model collaboration framework. To our knowledge, it is the first work to adopt such a collaborative paradigm for OCR hallucination mitigation. OCRAssistor incorporates minimal computational overhead, yet significantly improves LVLMs on both HalluText and OCRBench v2. Extensive experiments validate its effectiveness, efficiency, and scalability across a wide range of scenarios.

## 2 RELATED WORKS

### 2.1 OCR-AWARE BENCHMARK IN LVLM ERA

Before the LVLM era, OCR-aware benchmarks focused on specific sub-tasks, such as scene text detection and recognition (e.g., ICDAR (Karatzas et al., 2013), Total-Text (Ch'ng & Chan, 2017), SCUT-CTW1500 (Liu et al., 2017)), visual text understanding (e.g., TextVQA (Singh et al., 2019), STVQA (Biten et al., 2019)), key information extraction (e.g., FUNSD (Jaume et al., 2019), SROIE (Huang et al., 2019)), and chart understanding (e.g., ChartQA (Masry et al., 2022), infographicVQA (Mathew et al., 2022)). With the rise of LVLMs, the focus shifted towards unified OCR-centric benchmarks. OCRBench (Liu et al., 2024) integrates five major tasks—text recognition, scene VQA, document VQA, key information extraction, and handwritten formula recognition—across 27 datasets. The latest OCRBench v2 expands further, adding element parsing, knowledge reasoning, and mathematical calculations. Additionally, document parsing and understanding have gained widespread attention, with new benchmarks (Wei et al., 2024; Ouyang et al., 2025; Li et al., 2025) created for evaluating document-specific tasks. Recent research has also analyzed segmentation deficiencies, with OCR-Reasoning (Huang et al., 2025) and Reasoning OCR (He et al., 2025a) focusing on dense text understanding. Work by (Shu et al., 2025) and (He et al., 2025b) addresses hallucinations in non-semantic and occluded/blurred text-rich scenarios. In this paper, we propose a new benchmark to uncover OCR hallucinations in OCR-centric tasks, based on common failure cases from the general OCRBench v2.

### 2.2 HALLUCINATION MITIGATION

The concept of hallucination originates from the domains of pathology and psychology, where it is defined as the perception of something that does not exist in reality (Macpherson & Platchias, 2013). In natural language processing, hallucination typically refers to instances where generated content is implausible or inconsistent with the source input (Maynez et al., 2020). In the LVLM scenario, Hallucination refers to the phenomenon where the generated text response does not align with the corre-

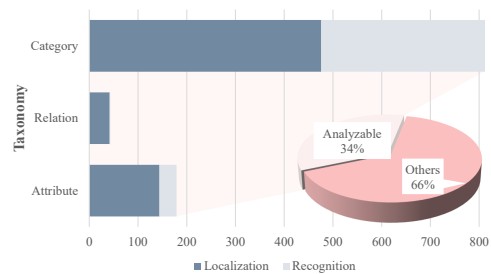

Figure 2: The distribution of failure cases on OCRBench v2.

sponding visual content (Bai et al., 2024). To address this, some methods improve training data (Liu et al., 2023; Yu et al., 2024a; Zhang et al., 2024), adopt architectural designs (Li et al., 2024b; Jain et al., 2024), or introduce post-training stages (Sun et al., 2023; Zhao et al., 2023; Gunjal et al., 2024). Given the high computational cost, others have explored training-free approaches (Leng et al., 2024; Wang et al., 2024c;a; Huang et al., 2024b; Favero et al., 2024; Zhu et al., 2025b), mainly categorized as contrastive decoding and attention intervention. Contrastive decoding (Leng et al., 2024; Wang et al., 2024c;a; Ghosh et al., 2025) modifies the decoding distribution but requires additional inference, leading to latency. Attention intervention (Huang et al., 2024b; Zhu et al., 2025b; Favero et al., 2024)shifts focus toward visual inputs during decoding but still incurs overhead. In OCR-centric tasks, we propose a large-small model collaboration framework that in-

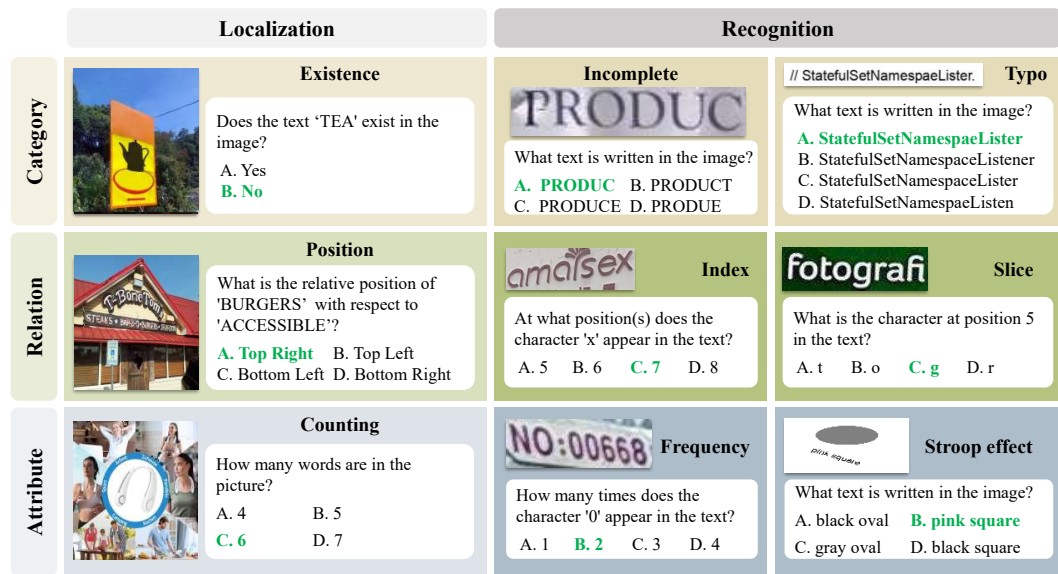

Figure 3: An overview of HalluText, which collects the challenging issues in visual text perception, including localization and recognition. The column axis involves the three categories of hallucination, *Category Hallucination*, *Relation Hallucination*, and *Attribute Hallucination*, respectively. Bold indicates the correct answer.

tegrates a lightweight, visually faithful OCR model into LVLMs to mitigate hallucination. This plug-and-play design improves visual alignment while avoiding the latency of prior training-free methods.

# 3 ANALYSIS ON VISUAL TEXT HALLUCINATION

## 3.1 EMPIRICAL ANALYSIS ON OCRBENCH V2

OCRBench v2 is currently the largest and most comprehensive benchmark for OCR-related tasks, comprising over 10,000 annotated question-answer pairs across more than 20 diverse scenarios in both Chinese and English. However, hallucinated samples are scattered across various task categories, making it challenging to systematically evaluate the hallucination-handling capabilities of LVLMs. To address this, we perform targeted analysis by identifying and categorizing hallucination types in model failures.

We select three representative LVLMs, including Qwen2.5-VL, MiniCPM-o 2.6, and MiniMonkey, and apply the official evaluation scripts to identify 3,006 samples where all models fail consistently. It is important to note that not all error cases are related to hallucinations. We consider only hallucination-related samples as analyzable. Other errors, such as those caused by question misinterpretation, complex reasoning failures, or annotation issues within the dataset, are not directly related to OCR hallucinations and are therefore excluded from our analysis.

To attribute hallucinations, we adopt two orthogonal dimensions: (1) perception task type (localization vs. recognition) and (2) hallucination type, which we refine for OCR settings as follows: **Category Hallucination**: incorrect recognition of text content or coordinates; **Relational Hallucination**: errors in spatial or semantic relations between text instances; **Attribute Hallucination**: incorrect description of text attributes such as quantity or color. Among the failed samples, 1,034 (34%) are deemed analyzable. We perform detailed attribution across models, and the resulting hallucination distributions, which are summarized in Figure 2, reveal consistent patterns across task types and models. These findings provide both conceptual grounding and empirical basis for developing robust hallucination benchmarks in OCR-focused LVLM evaluation.

Table 1: Distribution and original source of HalluText.

| Subsets | Existence | Position | Counting | Stroop | Typo | Incomplete | Freq. | Index | Slice |
|---|---|---|---|---|---|---|---|---|---|
| Number | 250 | 500 | 233 | 200 | 500 | 1495 | 500 | 500 | 500 |
| Source | SCUT-ENS (Liu et al., 2020) | Total-Text (Ch'ng & Chan, 2017) | ICDAR2013 (Karatzas et al., 2013) | Manual - | Typo-corpus (Hagiwara & Mita, 2019) | Union-Incomplete | Union-contextless (Jiang et al., 2023) | | |

## 3.2 HALLUTEXT BENCHMARK

Building on the empirical analysis in the previous section, we identify the distribution of hallucinated samples within OCRBench v2. Based on the occurrence scenarios of these hallucinations, we construct a dedicated dataset for OCR hallucination research, named HalluText, by reorganizing existing OCR datasets according to hallucination types. HalluText consists of 9 subsets, each corresponding to a specific hallucination category, and includes a total of 4,678 image–question–answer triplets. The definitions and construction procedures of each subset are detailed in the following section. The distribution and sources of the subsets are summarized in Table 1. The detailed construction procedures of all subsets are provided in Appendix B.

**Existence:** Due to training data biases, LVLMs are prone to hallucinations when presented with manipulated images. This subset is constructed from the scene text erasure dataset SCUT-ENS (Liu et al., 2020), with the goal of evaluating whether LVLMs can accurately perceive the presence of specific words in an image. To address the balance between *Yes* and *No* answers, we also incorporate negative polarity questions during the question construction process.

**Incompletion & Typo:** Influenced by the Language model, LVLMs tend to replace non-semantic words in their outputs with semantically plausible text. We constructed the Incomplete and Typo subsets using scene text that is affected by occlusion or truncation, and text containing common spelling errors, respectively. These subsets are designed to evaluate the ability the capability of accurately recognizing visual text while remaining robust to linguistic priors.

**Position:** Empirical studies reveal that LVLMs exhibit limitations in relative position perception. To evaluate their ability to understand spatial relationships in real-world scenes, we design a relative position recognition task based on scene text data. This task assesses how well LVLMs can perceive relative positions of visual elements across the entire image.

**Index & Slice:** Correspondingly, we also observe relation-level hallucinations within individual text instances. To minimize the influence of semantic priors, we construct position-specific questions on the Union14M-Contextless subset (Jiang et al., 2023), using common string slicing and indexing operations for naming. These subsets are designed to evaluate the ability to perceive intra-word spatial relations within single text instances.

**Counting:** Empirical results suggest that LVLMs struggle with counting-related tasks. To evaluate their ability to perceive numerical attributes of visual text, we construct a counting task based on the ICDAR2013 dataset, which primarily consists of focused text with minimal ambiguity. This subset is designed to assess whether LVLMs can accurately determine the number of text instances in an image.

**Frequency & Stroop Effect:** Beyond counting, color perception represents another key aspect of attribute-level hallucination. In addition to the intra-text counting task, we draw inspiration from the Stroop Effect (MacLeod, 1991) to construct synthetic images containing color and shape words. These two subsets are designed to evaluate the ability of LVLMs to suppress hallucinations related to text quantity and text color, respectively. More details are illustrated in Algorithm 1.

To ensure the quality of HalluText, we filtered out samples with blurred or occluded text and removed short, duplicate, or overly close text instances that could introduce spatial ambiguity. Each QA pair was then independently reviewed by two annotators and retained only when both agreed that the text was clearly readable, the visual evidence was sufficient, and the question–answer template was unambiguous, ensuring high inter-annotator reliability. Distractors were designed to be visually plausible yet falsifiable, using strict geometric rules for spatial relations and ±1 adjustments for counting tasks to maintain controlled difficulty. These steps ensure that HalluText isolates hallucination behavior without being confounded by image-quality issues or ambiguous annotations.

## 3.3 METRIC

Following OCRBench v2, we adopt a multiple-choice QA format with up to four options per question, using accuracy as the evaluation metric. For standardized evaluation, given images $\mathcal{I}$, questions $\mathcal{Q} = \{q_i\}_{i=1}^N$, and answers $\mathcal{A} = \{a_i\}_{i=1}^N$, we employ a fixed prompt template: "*Please strictly follow these rules: Only output the letter of the correct answer. Place the answer on a separate last line. Question: {question}. Answer:{}.*" The templatized questions are then fed into LVLMs to obtain predictions $\mathcal{P} = \{p_i\}_{i=1}^N = \mathcal{M}(\mathcal{I}, \mathcal{Q})$, where $\mathcal{M}$ is the LVLM. The multi-choice accuracy is formulated as $Acc = \frac{1}{N} \sum_{i=1}^N \mathbb{1}(p_i = a_i)$, where $\mathbb{1}$ is the indicator function.

## 4 METHOD

Inspired by (Ghosh et al., 2025), we introduce the OCRAssistor, an OCR-guided decoding framework that pioneers a collaborative mechanism between a large vision-language model (LVLM) and a lightweight OCR expert model to alleviate perception hallucination in OCR-centric tasks. The overall pipeline is illustrated in Figure 4.

Given an input image and a textual prompt, we first extract textual elements from the image using a lightweight OCR model, resulting in a sequence $X_{\text{OCR}} = \{o_1, o_2, \ldots, o_k\}$. To ground the LVLM's generation in these visual texts, we prepend them to the user prompt $X_{\text{prompt}} = \{x_1, x_2, \ldots, x_n\}$, yielding the augmented prompt $X_{\text{concat}} = \{o_1, \ldots, o_k, x_1, \ldots, x_n\}$.

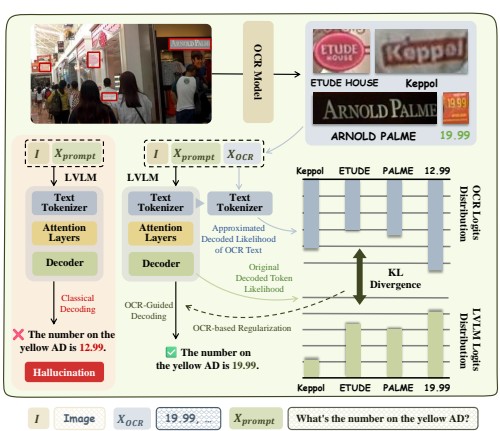

Figure 4: The framework of OCRAssistor.

Next, we construct a reference token distribution over the model's vocabulary $\mathcal{V}$ from the OCR outputs. Specifically, we pass the OCR text through the LVLM's embedding and output layers to obtain pseudo-logits $\hat{\ell}_{\text{OCR}} \in \mathbb{R}^{|\mathcal{V}|}$, which approximate the token likelihoods if the model were asked to generate the OCR text. These logits are normalized with a temperature-scaled softmax:

$$\hat{p}_{\text{OCR}}(w) = \frac{\exp(\hat{\ell}_{\text{OCR}}(w)/T)}{\sum_{v \in \mathcal{V}} \exp(\hat{\ell}_{\text{OCR}}(v)/T)}, \quad T > 0, \tag{1}$$

where $T$ controls the sharpness of the distribution. This step ensures that all tokens receive non-zero probability mass. During decoding, let $\mathcal{L}_i \in \mathbb{R}^{|\mathcal{V}|}$ denote the original decoded logits predicted by the LVLM $\mathcal{M}$ at step $i$ for the next token, and let $p_i(w) = \text{softmax}(\mathcal{L}_i)(w)$ be the corresponding token distribution. Motivated by KL-divergence (Kullback, 1951), we incorporate the OCR guidance by directly modifying the logits in a distribution-aware manner:

$$\mathcal{L}_i'(w) = \mathcal{L}_i(w) - \lambda \cdot \log \frac{p_i(w)}{\hat{p}_{\text{OCR}}(w)}, \quad \forall w \in \mathcal{V}, \tag{2}$$

where $\lambda$ is a hyperparameter controlling the strength of OCR-based regularization. To ensure the robustness of our design choices, we provide a detailed ablation on $\lambda$ in Table 13 of Appendix D. Intuitively, tokens more consistent with OCR-derived probabilities are relatively boosted, while inconsistent ones are penalized. The adjusted logits are then normalized to obtain the final decoding distribution:

$$p'(w \mid x_{<i}) = \text{softmax}(\mathcal{L}_i')(w). \tag{3}$$

At each step, the next token is sampled from $p'$, using the same decoding configuration as the base model. This ensures that the OCR guidance is seamlessly integrated into standard LVLM decoding while encouraging semantic consistency with visual texts. The generation terminates once an end-of-sequence token is produced or a predefined length limit is reached.

This approach integrates the predictions of an external OCR model into the decoding process via a KL-divergence-based guidance mechanism. This alignment encourages the LVLM to focus more

Table 2: Performance comparison on HalluText. OA indicates the OCRAssistor. Abbreviations: EX = Existence, RP = Relative Position, CT = Counting, ST = Stroop Effect, TY = Typo, IC = Incompletion, FQ = Frequency, ID = Index.

| Model | Localization | | | $Acc_{loc}$ | Recognition | | | | | | $Acc_{rec}$ | $Acc_{all}$ |
|---|---|---|---|---|---|---|---|---|---|---|---|---|
| | EX | POS | CT | | ST | TY | IC | FQ | ID | SL | | |
| *Proprietary LVLMs* | | | | | | | | | | | | |
| Gemeni-Pro | 91.6 | 63.8 | 69.1 | 74.8 | 99.0 | 66.8 | 56.1 | 63.7 | 46.2 | 62.8 | 65.8 | 68.8 |
| GPT-4o | 89.2 | 49.0 | 56.7 | 65.0 | 98.5 | 95.2 | 75.1 | 64.9 | 66.9 | 60.1 | 76.8 | 72.8 |
| *Open-source LVLMs* | | | | | | | | | | | | |
| Qwen2.5-VL-3B | 90.8 | 39.4 | 46.4 | 58.9 | 94.0 | 79.2 | 42.1 | 49.5 | 45.0 | 47.7 | 59.6 | 59.3 |
| Qwen2.5-VL-7B | 98.4 | 50.4 | 59.7 | 69.5 | 90.5 | 88.8 | 72.3 | 62.3 | 50.3 | 49.8 | 69.0 | 69.1 |
| Qwen2.5-VL-32B | 94.4 | 55.4 | 63.1 | 71.0 | 92.0 | 88.4 | 81.6 | 69.6 | 44.3 | 47.3 | 70.5 | 70.7 |
| InternVL3-2B | 75.2 | 31.8 | 44.2 | 50.4 | 99.0 | 82.8 | 62.6 | 45.6 | 38.4 | 40.7 | 61.5 | 57.8 |
| InternVL3-8B | 89.2 | 36.8 | 67.0 | 64.3 | 59.8 | 90.6 | 63.4 | 53.3 | 52.6 | 56.6 | 62.7 | 63.3 |
| InternVL3-14B | 95.6 | 67.2 | 63.1 | 75.3 | 99.5 | 92.8 | 88.1 | 75.7 | 60.0 | 63.8 | 80.0 | 78.4 |
| MiniCPM2.6-o-8B | 77.6 | 51.2 | 51.9 | 60.2 | 96.5 | 82.8 | 77.1 | 51.9 | 44.8 | 51.4 | 67.4 | 65.0 |
| MiniMonkey-2B | 76.4 | 32.4 | 35.2 | 48.0 | 76.9 | 71.0 | 77.1 | 43.8 | 6.0 | 24.4 | 49.9 | 49.2 |
| LLaVA-NeXT-7B | 74.0 | 38.6 | 31.3 | 48.0 | 71.4 | 48.6 | 41.5 | 64.3 | 51.8 | 39.6 | 44.6 | 45.7 |
| LLaVA-NeXT-7B + OA | 81.6 | 37.2 | 42.1 | 53.6 | 94.0 | 63.4 | 73.5 | 49.3 | 32.4 | 35.7 | 58.1 | 56.6 (+10.9) |
| Qwen2.5-VL-7B | 98.4 | 50.4 | 59.7 | 69.5 | 90.5 | 88.8 | 72.3 | 62.3 | 50.3 | 49.8 | 69.0 | 69.1 |
| Qwen2.5-VL-7B + OA | 98.8 | 50.4 | 66.5 | 71.9 | 95.5 | 89.8 | 81.3 | 71.0 | 72.9 | 82.0 | 82.1 | 78.7 (+9.6) |

heavily on visually grounded textual cues, effectively suppressing hallucinations and improving visual fidelity. Unlike prior comparison-based decoding methods, such as Visual Description Grounding Decoding (VDGD), which require multiple rounds of model inference, our approach achieves efficient inference by performing only one forward pass through the LVLM and a lightweight OCR model. This significantly reduces computational cost while maintaining strong performance.

## 5 EXPERIMENTS

### 5.1 SETTINGS

We evaluate and compare HalluText and OCRBench v2 with several state-of-the-art LVLMs, including proprietary models GPT-4o (Hurst et al., 2024) and Gemini-Pro (Team et al., 2024), as well as open-source models InternVL3 (Zhu et al., 2025a), Qwen2.5-VL (Bai et al., 2025), LLaVA-NeXT (Li et al., 2024a), MiniCPM2.6-o (Yao et al., 2024), and MiniMonkey (Huang et al., 2024a). We facilitate the widely used OCR engine PaddleOCR-v5 as our OCR model. To ensure fair comparison, we locally re-infer representative open-source LVLMs using only the annotated question prompts, and follow the official evaluation protocol. The maximum number of generated tokens is set to 1024. The temperature factor $T$ and regularization factor $\lambda$ are set to 0.1 and 0.1 by default. The detailed prompt settings are discussed in Appendix C.

### 5.2 RESULTS AND ANALYSIS

#### 5.2.1 HALLUTEXT

Table 2 presents the results on our proposed HalluText. We have several findings:

**1) OCR-centric hallucination remains an unsolved challenge across both proprietary and open-source models.** Overall, all models achieve less than 80% accuracy on our benchmark, and their performance on fine-grained subsets, such as relative position, counting, index, and slice, is significantly lower than the average accuracy $Acc_{all}$. This highlights the persistent and under-addressed issue of OCR hallucination in current LVLMs. The poor performance on *Slice* and *Index* suggests a limited understanding of ordinal relationships. *Counting* and *Relative position* tasks remain difficult due to the insensitivity of LVLMs to object-level correlation and the lack of fine-grained perceptual reasoning. Moreover, subsets like *Frequency* and *Slice*, which lack contextual information, expose the reliance of LVLMs on semantic cues for accurate recognition.

**2) The scaling law continues to hold for the OCR hallucination task.** We evaluate recent versions of Qwen and InternVL across three model scales and observe a consistent trend: larger models

Table 3: Performance comparison on OCRBench v2. OA indicates the OCRAssistor. Abbreviations: TR = Text Recognition, TD = Text Detection, TS = Text Spotting, RE = Relation Extraction, EP=Element Parsing, MC = Metathetical Calculating, TU=Text Understanding, KR = Knowledge Reasoning.

| Model | English Part | | | | | | | | Chinese Part | | | | | Overall | |
|---|---|---|---|---|---|---|---|---|---|---|---|---|---|---|---|
| | TR | TD | TS | RE | EP | MC | TU | KR | TR | RE | EP | TU | KR | English | Chinese |
| *Proprietary LVLMs* | | | | | | | | | | | | | | | |
| Gemini-Pro | 61.2 | 39.5 | 13.5 | 79.3 | 39.2 | 47.7 | 75.5 | 59.3 | 52.5 | 47.3 | 30.9 | 51.5 | 33.4 | 51.9 | 43.1 |
| GPT-4o | 61.2 | 26.7 | 0.0 | 77.5 | 36.3 | 43.4 | 71.1 | 55.5 | 21.6 | 53.0 | 29.8 | 38.5 | 18.2 | 46.5 | 32.2 |
| *Open-source LVLMs* | | | | | | | | | | | | | | | |
| MiniMonkey-2B | 58.1 | 19.6 | 0.0 | 51.3 | 33.0 | 15.7 | 61.7 | 44.8 | 61.4 | 40.5 | 27.9 | 42.8 | 17.9 | 35.5 | 38.1 |
| MiniCPM-o-2.6-8B | 67.4 | 26.5 | 0.0 | 70.1 | 34.0 | 31.7 | 70.6 | 57.6 | 54.7 | 52.4 | 27.6 | 42.5 | 31.6 | 44.8 | 41.7 |
| InternVL3-8B | 66.9 | 25.7 | 0.0 | 85.3 | 36.8 | 34.4 | 72.3 | 58.8 | 67.6 | 56.9 | 32.7 | 53.8 | 36.7 | 47.5 | 49.5 |
| LLaVA-NeXT-7B | 38.0 | 18.5 | 0.0 | 21.0 | 9.8 | 13.3 | 65.9 | 48.6 | 5.8 | 9.3 | 14.1 | 4.0 | 1.6 | 26.9 | 7.0 |
| LLaVA-NeXT+OA | 47.2 | 19.1 | 0.0 | 60.4 | 22.7 | 22.0 | 64.4 | 45.0 | 31.0 | 29.1 | 18.2 | 44.0 | 18.1 | 35.1 (+8.2) | 28.1 (+21.1) |
| Qwen2.5-VL-7B | 67.0 | 22.3 | 0.0 | 76.8 | 28.2 | 34.1 | 72.0 | 56.3 | 69.0 | 52.7 | 42.3 | 43.3 | 37.9 | 44.6 | 49.1 |
| Qwen2.5-VL-7B+OA | 60.4 | 22.6 | 0.0 | 86.4 | 33.6 | 46.2 | 72.9 | 54.7 | 57.0 | 64.8 | 39.4 | 56.8 | 45.8 | 47.1 (+2.5) | 52.8 (+3.7) |

Table 4: Performance comparison on different hallucination mitigating methods. OA indicates the OCRAssistor. Consensus are ensembled with Qwen2.5VL-3B, InternVL-2B and LLaVA-NeXT-7B. Abbreviations: EX = Existence, RP = Relative Position, CT = Counting, ST = Stroop Effect, TY = Typo, IC = Incompletion, FQ = Frequency, ID = Index.

| Model | Localization | | | $Acc_{loc}$ | Recognition | | | | | | $Acc_{rec}$ | $Acc_{all}$ |
|---|---|---|---|---|---|---|---|---|---|---|---|---|
| | EX | POS | CT | | ST | TY | IC | FQ | ID | SL | | |
| SemanticHallu | 90.0 | 40.0 | 49.1 | 59.7 | 97.0 | 74.2 | 45.6 | 53.2 | 51.9 | 46.3 | 61.4 | 60.8 |
| Consensus | 83.6 | 36.0 | 44.6 | 54.7 | 86.9 | 69.4 | 42.3 | 44.5 | 41.6 | 46.6 | 55.2 | 55.1 |
| VDGD | 94.4 | 39.6 | 49.8 | 61.3 | 96.5 | 78.2 | 74.8 | 47.9 | 47.6 | 43.2 | 64.7 | 63.6 |
| OCRAssistor | **96.4** | **40.6** | **50.6** | **62.5** | **100.0** | **83.8** | **76.2** | 41.6 | 44.4 | 43.0 | **64.8** | **64.1** |

exhibit stronger capabilities in suppressing textual hallucinations, confirming the applicability of scaling effects in this domain.

**3) Our OCRAssistor method, under a training-free setting, integrates an off-the-shelf open-source OCR model and yields substantial improvements.** Specifically, it improves LLaVA by 10.9% and Qwen2.5-VL by 9.6%, with consistent gains across nearly all fine-grained subsets. These results demonstrate the effectiveness of our approach in mitigating OCR hallucinations.

### 5.2.2 OCRBENCH V2

We further evaluate our approach on OCRBench v2, a general benchmark for OCR-centric tasks. Table 3 shows that our method improves LLaVA-NeXt-7B by 8.2% in English scenarios and 21.1% in Chinese scenarios. For Qwen2.5-VL-7B, which possesses stronger baseline capabilities, our method still achieves 2.5% and 3.7% improvements in English and Chinese settings, respectively. The notably larger gain in LLaVA's Chinese performance is primarily due to the relatively limited Chinese data exposure during its pretraining phase, compared to Qwen2.5-VL. This suggests that our method can effectively compensate for underrepresented modalities or languages in pretraining, particularly in low-resource scenarios. Beyond perception tasks, OCRAssistor also improves relation extraction, text comprehension, and knowledge reasoning. These results show that integrating an OCR model not only benefits visual-text perception tasks but also enhances high-level semantic understanding in LVLMs.

### 5.2.3 COMPARISON OF OTHER HALLUCINATION MITIGATION METHODS

We compared OCRAssistor with three representative hallucination-mitigation approaches: (1) a visual-attention–enhanced variant (Shu et al., 2025), (2) a consensus-based ensemble method (Zhang et al., 2025), and (3) the contrastive decoding strategy VDGD (Ghosh et al., 2025). As shown in the Table 4, OCRAssistor achieves the best performance across all HalluText subsets, surpassing the consensus-based method by +9.8% and VDGD by +1.3%, while also being substantially more efficient. These gains mainly come from incorporating reliable external OCR evidence, which provides explicit grounding for both detection and recognition. In contrast, uncertainty-based

Table 5: Ablation for all components. Baseline selects Qwen2.5-VL-7B. CoT means the prompt includes the chain-of-thought instruction "*Let us think this question step by step.*". OCR means the OCR results are included in the prompt directly. OCRAssistor (OA) is our final version. OA indicates the OCRAssistor. Abbreviations: EX = Existence, RP = Relative Position, CT = Counting, ST = Stroop Effect, TY = Typo, IC = Incompletion, FQ = Frequency, ID = Index.

| Model | Localization | | | $Acc_{loc}$ | Recognition | | | | | | $Acc_{rec}$ | $Acc_{all}$ |
|---|---|---|---|---|---|---|---|---|---|---|---|---|
| | EX | POS | CT | | ST | TY | IC | FQ | ID | SL | | |
| Qwen2.5VL-7B | 98.4 | 50.4 | 59.7 | 69.5 | 90.5 | 88.8 | 72.3 | 62.3 | 50.3 | 49.8 | 69.0 | 69.2 |
| Qwen2.5VL-7B + CoT | 98.4 | 50.0 | 63.1 | 70.5 | 92.0 | 88.2 | 56.2 | 65.1 | 77.6 | 80.6 | 76.6 | 74.6 |
| Qwen2.5VL-7B + OCR | 98.0 | 52.2 | 60.1 | 70.1 | 99.0 | 90.0 | 89.4 | 56.4 | 48.7 | 49.0 | 72.1 | 71.4 |
| Qwen2.5VL-7B + OCR + CoT | 98.0 | 48.6 | 69.1 | 71.9 | 95.0 | 89.6 | 63.5 | 64.7 | 74.7 | 80.2 | 78.0 | 75.9 |
| Qwen2.5VL-7B + OA | **98.8** | 50.4 | 66.5 | **71.9** | 95.5 | 89.8 | 81.3 | **71.0** | 72.9 | **82.0** | **82.1** | **78.7** |

Table 6: Performance comparison on ChartQA and DocVQA. OA indicates our method OCRAssistor.

| Dataset | Base | Base + OA |
|---|---|---|
| ChartQA | 69.1 | **70.4** |
| DocVQA | 49.3 | **57.1** |

Table 7: Effect on OCR quality. OA indicates the OCRAssistor. According to verification of ground truth, samples are divided into Correct and Incorrect categories.

| Settings | ST | TY | IC | FQ | ID | SL | $Acc_{rec}$ |
|---|---|---|---|---|---|---|---|
| Qwen2.5VL-3B+OA | 100 | 83.8 | 76.2 | 41.6 | 44.4 | 43.0 | 64.8 |
| Correct OCR | 100 | 85.4 | 84.4 | 74.2 | 77.8 | 86.2 | 84.7 |
| Incorrect OCR | 100 | 75.7 | 61.5 | 40.9 | 42.2 | 46.0 | 61.1 |

methods fail to correct visual–text perception errors, especially in cases of mis-detection, relational confusion, or attribute mistakes.

### 5.2.4 ABLATIONS

In this section, we conduct a series of ablation studies under different experimental configurations to investigate which components contribute most to reducing hallucination in LVLMs, shown in Table 5. Specifically, we compare the following setups on the HalluText benchmark: (1) adding chain-of-thought (CoT) prompting, (2) simply appending raw OCR outputs to the prompt, and (3) our proposed OCRAssistor strategy. Results show that directly appending OCR results to the prompt brings modest gains on average. While CoT prompting yields some improvement, our OCRAssistor demonstrates substantially stronger gains, achieving improvements of 2.4% in localization, 13.1% in recognition, and 9.6% on the overall average metric. These results confirm that our carefully designed OCRAssistor effectively and seamlessly integrates OCR information into LVLMs. By leveraging the structured visual guidance provided by the OCR model, our method significantly alleviates OCR-aware hallucinations in both perception and understanding tasks.

### 5.2.5 EXPERIMENTS ON OTHER TEXTVQA BENCHMARK

We additionally evaluated our method on two widely used open-ended benchmarks, ChartQA (Masry et al., 2022) and DocVQA (Mathew et al., 2021), to further assess its generalization beyond multiple-choice settings. Table 6 shows that OCRAssistor yields consistent improvements on both benchmarks. The gain on ChartQA is relatively smaller, which we attribute to the fact that the dataset involves substantial numerical reasoning beyond text recognition. Our method is designed to enhance the LVLM's ability to accurately perceive and ground textual content in images, thereby reducing hallucinations arising from internal priors. Although OCRAssistor does not explicitly target complex reasoning, the improved visual-text grounding still contributes to measurable performance improvements on both ChartQA and DocVQA.

### 5.2.6 OCR QUALITY

To directly assess this failure mode, we conducted a controlled experiment simulating the "OCR failure" scenario. We partitioned the evaluation samples into two subsets according to the correctness of the OCR output: a Correct set, where all OCR-extracted text matches the ground-truth annota-

Table 8: The efficiency experiments between OCRAssistor and VDGD (Ghosh et al., 2025). For convenience, we use Qwen2.5-VL-3B as the base model.

| Settings | HalluText | Time(s/image) |
|---|---|---|
| Qwen2.5-VL-3B | 59.3 | 0.275 |
| Qwen2.5-VL-3B + VDGD | 63.6 | 10.972 |
| Qwen2.5-VL-3B + OA | 64.1 | 0.817 |

Table 9: The gains of OCRAssistor on Qwen2.5-VL series across different scales.

| Model | HalluText | OCRBench v2(EN/ZH) |
|---|---|---|
| Qwen2.5-VL-3B | +4.8 | +1.3 / +4.7 |
| Qwen2.5-VL-7B | +9.6 | +2.5 / +3.7 |

tions, and an Incorrect set, where the OCR model produces mismatched or erroneous text. Results of Table 7 show that the Correct subset achieves an $Acc_{rec}$ of 84.7, reflecting the upper bound of the benefit when OCR cues are fully accurate. The Incorrect subset yields an $Acc_{rec}$ of 61.1, confirming that incorrect OCR cues do introduce noise and can degrade performance. Crucially, accuracy on the Incorrect subset remains far above naive baselines (e.g., random choice or the LVLM without assistance), demonstrating that OCRAssistor does not blindly follow faulty OCR outputs. Instead, its soft-guidance formulation allows the LVLM to partially resist or correct misleading cues, indicating a non-trivial degree of robustness even under simulated OCR failure.

### 5.2.7 EFFICIENCY

Table 8 presents the runtime performance of OCRAssistor on Qwen2.5-VL. We compare three setups: the original 3B model, the VDGD-enhanced model, and our OCRAssistor. OCRAssistor introduces only 0.6s of additional latency per image while delivering a 4.8% performance gain. In contrast, VDGD adds over 10s per image, making it impractical despite modest improvements. This demonstrates OCRAssistor's favorable balance of efficiency and effectiveness. Notably, since our evaluation involves only multiple-choice outputs, baseline inference times remain low. In more complex scenarios requiring free-form or subjective generation, the overall inference latency would increase significantly, thereby reducing the relative overhead introduced by the OCR module. Thus, the efficiency advantage of our method could be more pronounced in real-world applications. Section F will discuss the efficiency of OCRAssistor under a more open-ended generation scenario further.

### 5.2.8 SCALING

We further examine the effectiveness of OCRAssistor across different model scales, with results summarized in Table 9. Experimental results demonstrate that OCRAssistor consistently yields performance gains across both model sizes. Notably, the improvements are more pronounced on HalluText, which is explicitly designed to evaluate hallucination, indicating that our decoding strategy is particularly effective in hallucination-prone scenarios. These findings highlight the robust generalization ability of our method across LVLMs of varying capacity, making it applicable to both lightweight and large-scale models.

## 6 CONCLUSION

In this work, we present a comprehensive study on OCR-centric hallucinations in LVLMs. After applying a dual-perspective taxonomy that categorizes errors by task process (localization, recognition) and hallucination type (category, relation, attribute) and analyzing failure cases, we introduce HalluText, a fine-grained benchmark comprising 4,678 samples across 9 subsets, designed to diagnose OCR hallucinations. To address these challenges, we develop OCRAssistor, a training-free and plug-and-play pipeline that leverages external OCR signals to guide LVLM decoding. Experiments on HalluText and OCRBench v2 show that OCRAssistor consistently improves performance across models of different scales, while remaining efficient and scalable. Our findings underscore not only the importance of structured OCR integration but also highlight the effectiveness of a large-small model collaboration paradigm, where a lightweight OCR expert module supplements the strengths of a powerful LVLM. This cooperative design offers a practical and generalizable solution for reducing hallucinations in vision-language understanding.

REPRODUCIBILITY STATEMENT

We have made extensive efforts to ensure that the results reported in this work are reproducible. All model architectures, training procedures, and hyperparameter settings are described in the main text (Sections 4–5) and detailed further in the Appendix (Appendix A–C). For the datasets used in our experiments, we provide complete descriptions of preprocessing and filtering steps in the supplementary materials. All evaluation metrics are formally defined in Section 3.3, enabling consistent replication of our analysis. Additionally, the source code and scripts used for training, inference, and evaluation will be made publicly available as anonymized supplementary material, facilitating direct reproduction of the reported results. Readers are referred to these resources for all necessary details to reproduce the experiments and analyses presented in this work.

ETHICS STATEMENT

All authors have read and adhered to the ICLR Code of Ethics. This work focuses on analyzing and mitigating OCR hallucination, and does not involve direct experimentation on human subjects. All datasets used are either publicly available or used under appropriate licenses, and any personal information has been anonymized to protect privacy. We are aware of potential societal impacts of multimodal AI systems, including misuse for generating misleading content or biased outputs. In our experiments, we take care to evaluate model behavior across diverse languages and scenarios to mitigate unintended bias. No datasets or methods used are expected to cause harm to individuals or communities. We encourage responsible use and recommend that future users of the proposed models follow relevant legal, privacy, and fairness guidelines. Any conflicts of interest have been disclosed, and all research practices adhere to established standards of scientific integrity.

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

The appendix includes the following aspects:

- A: Use of Large Language Models
- B: Details of HalluText curation.
- C: Details of the prompt.
- D: Additional Experiments.
- E: Visualization.
- F: Efficiency Analysis.
- G: OCR Quality.

## A  USE OF LARGE LANGUAGE MODELS

In this work, large language models (LLMs) are used solely as generally purpose assistive tools to improve the clarity, grammar, and readability of the manuscript. LLMs are not used for research ideation, data analysis, model development, or any other scientific decision-making. All scientific content, ideas, results, and conclusions presented in this paper are independently produced by the authors. The authors take full responsibility for the accuracy and integrity of the work, including any content that was refined or edited with the assistance of LLMs. No information generated by LLMs that could constitute plagiarism, fabrication, or scientific misconduct has been included.

## B  DETAILS OF HALLUTEXT CURATION

---

**Algorithm 1** Generating Stroop-Effect QA Pairs

---

**Require:** colors, shapes
**Ensure:** qa_pair
  1: **function** GENERATESTROOPQA
  2:     shape_color, text_color, render_color ← RandomSelect(colors)
  3:     shape_shape, text_shape ← RandomSelect(shapes)
  4:     img ← ImageDraw(shape_shape, shape_color, render_color)
  5:     text ← concatenate(text_color, text_shape)
  6:     font_size ← RandomSelect(range(10, 50))
  7:     bbox ← ComputeBBox(text, font_size)
  8:     **if** apply_rotation **then**
  9:        rotated_dims ← GetRotatedDims(bbox, angle)
 10:        **if** rotated_dims exceeds image boundaries **then**
 11:           font_size ← AdjustFontSize(font_size, max_scale)
 12:           bbox ← ComputeBBox(text, font_size)
 13:        **end if**
 14:        pos ← FindValidPos(rotated_dims)
 15:        RenderTextRotated(text, pos, font_size, angle)
 16:     **else**
 17:        pos ← FindValidPos(bbox)
 18:        RenderText(text, pos, font_size)
 19:     **end if**
 20:     question ← "What text is written in the image?"
 21:     options ← GenerateOptions(text_color, text_shape, shape_color, shape_shape, render_color)
 22:     options, answer ← ShuffleOptions(text, optiobs)
 23:     qa_pair ← {question, img, options, answer}
 24: **end function**

---

### B.1  EXISTENCE

We construct the Existence subset using SCUT-ENS Liu et al. (2020), a dataset containing paired images before and after scene text erasing. By leveraging these image pairs and their corresponding

Table 10: Prompt templates for different settings.

| Settings | Prompt |
|---|---|
| **HalluText** | |
| Baseline | Please strictly follow these rules: \n Place the answer only option letter (with no extra characters) on a separate last line. \n Question: [QUESTION]. \n Options: [OPTIONS]. \n Answer: \n |
| Baseline+CoT | Please strictly follow these rules: \n Let us think this question step by step (Chain of thought) and Place the answer only option letter (with no extra characters) on a separate last line. \n Question: [QUESTION]. \n Options: [OPTIONS]. \n Chain of thought: \n Answer: \n |
| Baseline+OCR | The texts in the image were recognized in the image: [OCR RESULTS] \Please strictly follow these rules: \n Place the answer only option letter (with no extra characters) on a separate last line. \n Question: [QUESTION]. \n Options: [OPTIONS].\n Answer: \n |
| Baseline+OCR+CoT | The texts in the image were recognized in the image: [OCR RESULTS]. Please strictly follow these rules: \n Let us think this question step by step (Chain of thought) and place the answer only option letter (with no extra characters) on a separate last line. \n Question: [QUESTION]. \n Options: [OPTIONS]. \n Chain of thought: \n Answer: \n |
| Baseline+OCRAssistor | Same as Baseline+OCR+CoT |

OCR annotations, we create VQA-style samples that ask whether a specific text instance existed prior to erasure. As shown in Figure 5, we design the questions template *"Does the text 'TEA' exist in the image?"*, accompanied by the erased image as shown in Figure 5 (a). The correct answer is clearly *No*. If the LVLM relies solely on dataset bias rather than visual information provided by the user, it is prone to incorrectly predicting *Yes*. To further balance the distribution of answers, we deliberately incorporate negative forms in the question design, ensuring a more even ratio between *Yes* and *No* responses.

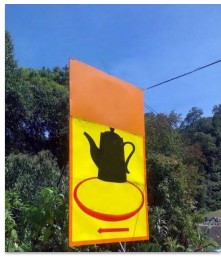 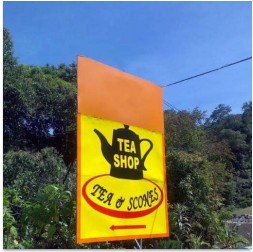

(a) Erased image      (b) Original image

Figure 5: The details of "Existence" subset.

## B.2 INCOMPLETE

The *Incompletion* subset is adapted from OCRBench v2 using its standard recognition prompts. To ensure quality, we manually verify and clean the original annotations. For each question, a confusion option is generated by using the original word, which is rich in semantics. The other two distractors are created by applying random character-level edits (insertion, deletion, substitution) based on the ground truth. An example is shown in Figure 3.

### B.3 COUNTING

The *Counting* subset also follows the instruction of OCRBench v2, using standard counting-style prompts. Answers are derived from the original dataset, with confusion options introduced by sampling positive integer near the correct answer (e.g. $+1$, $-1$) to simulate realistic ambiguity.

### B.4 TYPO

The *Typo* subset is synthetically constructed using the typo corpus and the Pillow image library. We adopt a minimalist rendering black text on a white background, without any decorative elements. Confusion options are generated by randomly applying one character-level edit (insertion, deletion, or substitution) to the ground truth or its corrected version. This process mirrors that of the Incomplete subset, focusing on recognition robustness under typographical noise.

### B.5 POSITION

The *Position* subset is built from the Total-Text dataset. We discard illegible or unannotated text instances and classify the remaining ones into eight relative positional categories: top-left, top, top-right, right, bottom-right, bottom, bottom-left, and left. To avoid ambiguity between adjacent classes (e.g., top-left vs. top), we explicitly remove potentially confusing categories when generating answer options. This ensures that each question has one unambiguous correct answer.

### B.6 INDEX, SLICE, AND FREQUENCY

The *Index*, *Slice*, and *Frequency* subsets are jointly derived from the Union14M-contextless dataset Jiang et al. (2023). For each image–annotation pair, we sample character-level statistics such as frequency, position, and index to formulate distinct question types. To maintain a single-answer format, we filter out ambiguous cases—such as words with repeated characters—where multiple valid answers might exist for an index-based query.

### B.7 STROOP EFFECT

The *Stroop Effect* subset is uniquely constructed without relying on any existing public dataset. We manually generate image–question–answer triplets to simulate conditions where irrelevant but plausible distractors interfere with OCR perception. The generation pipeline is detailed in Algorithm 1. Crucially, all confusion options used in the answer choices are explicitly present within the image, enabling a faithful evaluation of the LVLM's susceptibility to OCR hallucinations.

## C DETAILS OF PROMPT

To adapt different ablation settings, we design prompt templates tailored to various input configurations. The detailed prompt formats are provided in Table 10. In the HalluText benchmark, the *Baseline* configuration includes only the core question and the corresponding answer choices. For the *CoT* and *OCR* settings, we incorporate respective guiding cues into the prompt. In the OCRAssistor setup, we include both the CoT prompt and OCR information in the language instruction to encourage alignment between the LVLM's output and the OCR-derived content distribution. Experimental results show that our method significantly improves performance on HalluText and has a consistent positive effect in mitigating OCR hallucinations. On the OCRBench v2 benchmark, we follow the standard evaluation protocol, using only the question as the full prompt. Under the fair setting, our method demonstrates stable and consistent improvements in fair comparisons with other models, as shown in Table 3.

## D ADDITIONAL EXPERIMENTS

In this section, we provide additional experimental results that are omitted from the main text due to space limitations.

Table 11: Ablation for OCR inputs. The baseline LVLM is Qwen2.5-VL-3B. All settings loads OCRAssistor.

| Model | $Acc_{loc}$ | $Acc_{rec}$ | $Acc_{all}$ |
|---|---|---|---|
| Rec-only | 62.5 | 64.8 | 64.1 |
| Det & Rec | 63.5 (+1.0) | 62.5 (-2.3) | 62.8 (-1.3) |

Table 12: Ablation for the setting of $\lambda$, The baseline LVLM is Qwen2.5-VL-3B. **Bold** indicates the best performance.

| $\lambda$ | $Acc_{loc}$ | $Acc_{rec}$ | $Acc_{all}$ |
|---|---|---|---|
| 0.1 | **60.3** | 65.8 | **64.0** |
| 0.5 | 59.7 | **66.0** | 63.9 |
| 1.0 | 59.1 | 65.9 | 63.6 |
| 1.5 | 59.5 | 65.6 | 63.6 |
| 2.0 | 59.5 | 65.9 | 63.8 |

Table 13: Ablation for the setting of $T$, The baseline LVLM is Qwen2.5-VL-3B. **Bold** indicates the best performance.

| $T$ | $Acc_{loc}$ | $Acc_{rec}$ | $Acc_{all}$ |
|---|---|---|---|
| 0.1 | **62.5** | 64.8 | **64.1** |
| 0.5 | 61.0 | 65.2 | 63.8 |
| 1.0 | 60.3 | **65.8** | 64.0 |

## D.1 OCR INPUTS

We conduct an ablation study on the use of OCR inputs. Two configurations are compared: (1) using only the OCR recognition results as input, and (2) incorporating both detection and recognition results into the prompt. As shown in Table 11, providing both detection and recognition results as OCR priors leads to a 1.0% improvement on the localization task compared to using recognition results alone. However, this setting results in performance drops of 2.3% and 1.3% on the recognition task and the overall average, respectively. We attribute this phenomenon to the limited guidance provided by the coordinate-format detection results after tokenization, which could not be effectively utilized during LVLM decoding. We caution that directly including OCR detection outputs in the prompt make adverse effects.

## D.2 THE EFFECT OF $\lambda$ AND $T$

We conducted a comprehensive ablation study to evaluate the sensitivity of OCRAssistor to its two key hyperparameters: the guidance weight $\lambda$ and the temperature T. As shown in Tables ($\lambda$) and (T), the overall performance remains remarkably stable across a wide range of values. For $\lambda$, varying the weight from 0.1 to 2.0 results in only minimal fluctuations in both $Acc_{loc}$ and $Acc_{rec}$ (within ±0.3 on overall accuracy). Similarly, adjusting T from 0.1 to 1.0 produces highly consistent results, with the overall accuracy differing by less than 0.3 across settings. Importantly, no monotonic degradation or sharp peak is observed, indicating that OCRAssistor is not sensitive to either $\lambda$ or temperature, and the guidance effect remains robust under different strengths of modulation. Given this stability, we adopt $\lambda = 0.1$ and T = 0.1 in our main experiments.

## D.3 THE DETAILED RESULTS ON QWEN2.5-VL-3B

Owing to space constraints, Table 9 presents only the performance gains of Qwen2.5-VL-3B with OCRAssistor on HalluText and OCRBench v2. For completeness, the detailed results are provided in Table 14 and Table 15.

# E VISUALIZATION

This section presents qualitative visualizations of Qwen2.5-VL-7B's performance on two datasets.

Table 14: Detailed results of Qwen2.5-VL-3B on OCRBench v2. OA indicates the OCRAssistor. Abbreviations: TR = Text Recognition, TD = Text Detection, TS = Text Spotting, RE = Relation Extraction, EP=Element Parsing, MC = Metathetical Calculating, TU=Text Understanding, KR = Knowledge Reasoning.

| Model | English Part | | | | | | | | Chinese Part | | | | | Overall | |
| | TR | TD | TS | RE | EP | MC | TU | KR | TR | RE | EP | TU | KR | English | Chinese |
|---|---|---|---|---|---|---|---|---|---|---|---|---|---|---|---|
| Qwen2.5-VL-3B | 63.9 | 18.7 | 0.0 | 81.5 | 32.5 | 35.3 | 69.2 | 49.2 | 69.0 | 47.2 | 33.0 | 35.5 | 43.5 | 43.8 | 45.6 |
| Qwen2.5-VL-3B+OA | 58.9 | 20.6 | 0.0 | 84.7 | 34.6 | 39.9 | 70.9 | 51.0 | 67.6 | 54.8 | 33.2 | 54.0 | 41.8 | 45.1 (+1.3) | 50.3 (+4.7) |

Table 15: Detailed performance of Qwen2.5VL-3B on HalluText. OA indicates the OCRAssistor. Abbreviations: EX = Existence, RP = Relative Position, CT = Counting, ST = Stroop Effect, TY = Typo, IC = Incompletion, FQ = Frequency, ID = Index.

| Model | Localization | | | $Acc_{loc}$ | Recognition | | | | | | $Acc_{rec}$ | $Acc_{all}$ |
| | EX | POS | CT | | ST | TY | IC | FQ | ID | SL | | |
|---|---|---|---|---|---|---|---|---|---|---|---|---|
| Qwen2.5-VL-3B | 90.8 | 39.4 | 46.4 | 58.9 | 94.0 | 79.2 | 42.1 | 49.5 | 45.0 | 47.7 | 59.6 | 59.3 |
| Qwen2.5-VL-3B + OA | 96.4 | 40.6 | 50.6 | 62.5 | 100.0 | 83.8 | 78.2 | 41.6 | 44.4 | 43.0 | 64.8 | 64.1 |

### E.1  HALLUTEXT

Figure 6 illustrates results on HalluText, where we visualize the input image, question, answer options, model predictions (before and after enhancement), and the OCR-recognized text. The comparison shows that with guidance from OCR outputs, Qwen2.5-VL-7B better adheres to visual instructions and exhibits reduced OCR hallucinations.

### E.2  OCRBENCH V2

Figure 7 and Figure 8 show visualizations on the English and Chinese subsets of OCRBench, respectively. We observe that the proposed OCRAssistor module helps the LVLM correct fine-grained recognition errors. For example, in Figure 7, the model originally extracted *"Newspaper Parent"* for the field *"Brand(s) Applicable"*, while the image text actually reads *"Newport Parent"*; similarly, it misread *"Coupon Issue Date"* as *"4/1/00"* instead of the correct *"4/14/00"*. These cases highlight the presence of OCR hallucinations in the baseline LVLM, which are significantly mitigated after applying the proposed improvements. In summary, our method achieves stable performance gains across diverse generalized OCR scenarios.

## F  EFFICIENCY ANALYSIS

As a supplement to Table 8, we further evaluate the efficiency of OCRAssistor on OCRBenchv2, an open-ended generation benchmark in Table 16. The results show that OCRAssistor consistently achieves the highest overall accuracy across both the 128- and 1024-token settings (e.g., TR, RE, TU), while maintaining substantially lower computational overhead. At 128 tokens, OCRAssistor requires only 1.12 seconds, 3.3× faster than TextHallu and 8× faster than VDGD. Even at 1024 tokens, its runtime remains low (1.24 seconds), still achieving 3× and 9× speedups over TextHallu and VDGD, respectively. Regarding scalability, OCRAssistor exhibits only an 11% increase in latency from 128 to 1024 tokens, closely matching the baseline trend and outperforming both VDGD's steep growth and the consistently high cost of TextHallu. Overall, OCRAssistor provides the best accuracy–efficiency trade-off: it delivers higher performance across key metrics while adding only less than 1 second of overhead, and preserves robust scalability for longer input sequences.

## G  OCR QUALITY

We also investigate the impact of OCR quality on hallucination mitigation, as shown in Table 17. Specifically, we evaluate the recognition quality of OCR models on the 1,500 original images used

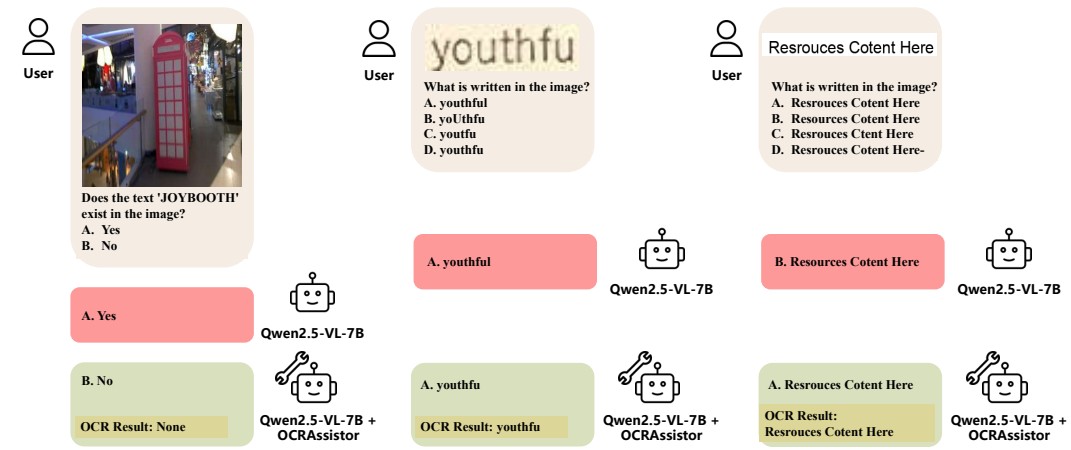

Figure 6: A Visualization of HalluText.

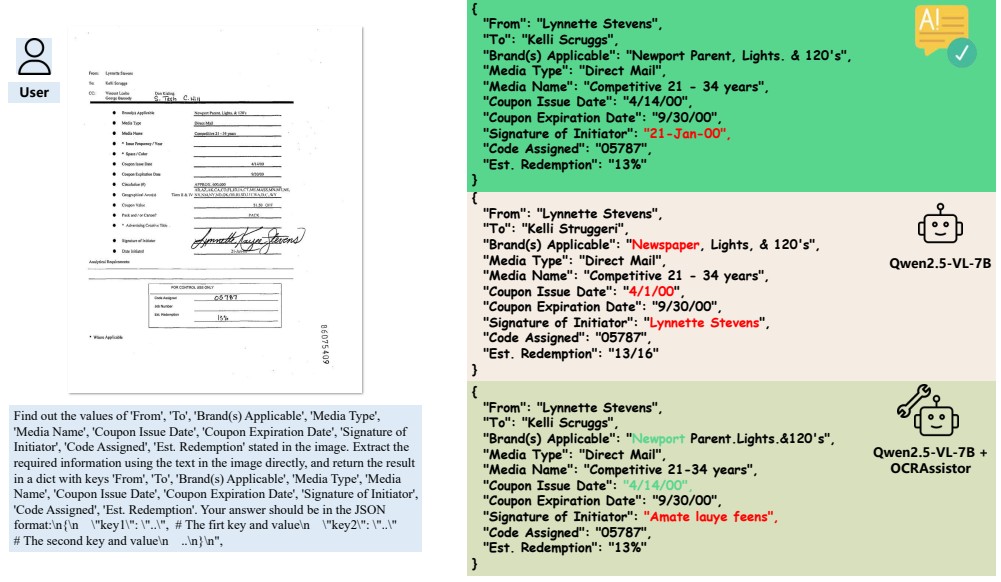

Figure 7: A Visualization of OCRBench v2-EN.

to construct the HalluText benchmark. In addition to our default OCR system PaddleOCR[1], we compare with another widely used alternative, EasyOCR[2]. Experimental results indicate that EasyOCR achieves a 1-N.E.D. score that is 2.2 points lower than PaddleOCR, suggesting slightly inferior recognition performance. Correspondingly, under the same experimental settings, the downstream results on HalluText using EasyOCR are consistently lower than those with PaddleOCR. These findings demonstrate a positive correlation between OCR quality and hallucination-mitigating performance: higher-quality OCR outputs provide more reliable visual cues, which better guide LVLMs and reduce hallucinated generations.

---

[1] https://github.com/PaddlePaddle/PaddleOCR
[2] https://github.com/JaidedAI/EasyOCR

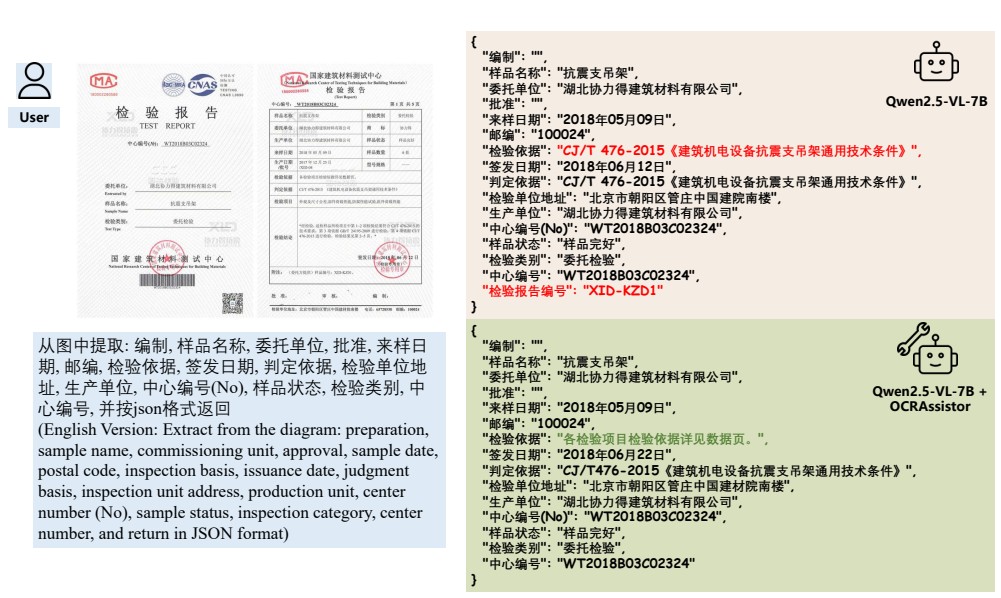

Figure 8: A Visualization of OCRBench v2-CN.

Table 16: Perfomance comparison on OCRBench v2. OA indicates the OCRAssistor. Abbreviations: TR = Text Recognition, TD = Text Detection, TS = Text Spotting, RE = Relation Extraction, EP=Element Parsing, MC = Metathetical Calculating, TU=Text Understanding, KR = Knowledge Reasoning.

| Model | English Part | | | | | | | | Chinese Part | | | | | Overall | | Time |
|---|---|---|---|---|---|---|---|---|---|---|---|---|---|---|---|---|
| | TR | TD | TS | RE | EP | MC | TU | KR | TR | RE | EP | TU | KR | English | Chinese | |
| Baseline-128 | 57.9 | 18.6 | 0 | 67.9 | 15.6 | 33.4 | 69.1 | 47.9 | 41.2 | 46.2 | 23.2 | 37.0 | 43.1 | 38.8 | 38.1 | 0.66 |
| Baseline-1024 | 64.1 | 18.6 | 0 | 81.3 | 32.5 | 35.3 | 69.0 | 49.0 | 69.0 | 47.3 | 32.9 | 36.5 | 43.1 | 43.7 | 45.8 | 0.96 |
| TextHalllu-128 | 58.9 | 20.9 | 0 | 67.8 | 14.7 | 31.7 | 64.9 | 48.2 | 37.5 | 32.3 | 21.0 | 29.8 | 42.5 | 38.4 | 32.6 | 3.68 |
| TextHalllu-1024 | 66.4 | 20.9 | 0 | 81.5 | 30.3 | 35.7 | 64.9 | 49.3 | 67.8 | 34.1 | 30.7 | 32.8 | 42.0 | 43.6 | 41.5 | 3.60 |
| VDGD-128 | 53.6 | 20.2 | 0 | 64.1 | 16.8 | 28.5 | 69.6 | 47.0 | 33.9 | 46.4 | 25.1 | 28.6 | 38.5 | 37.5 | 34.5 | 8.97 |
| VDGD-1024 | 60.7 | 20.3 | 0 | 78.8 | 34.0 | 31.7 | 69.6 | 48.1 | 62.0 | 47.8 | 33.7 | 31.0 | 38.6 | 42.9 | 42.6 | 10.95 |
| OCRAssistor-128 | 52.9 | 21.5 | 0 | 70.4 | 17.0 | 34.6 | 71.3 | 49.2 | 40.0 | 53.9 | 25.3 | 57.0 | 42.4 | 39.6 | 43.7 | 1.12 |
| OCRAssistor-1024 | 59.4 | 21.4 | 0 | 83.9 | 34.6 | 38.9 | 71.2 | 50.4 | 69.9 | 55.1 | 31.5 | 56.0 | 42.9 | 45.0 | 51.1 | 1.24 |

Table 17: Effect of different OCR models. 1-N.E.D. is a recognition metric defined as $1 - NED$.

| OCR Model | 1-N.E.D. | $Acc_{loc}$ | $Acc_{rec}$ | $Acc_{all}$ |
|---|---|---|---|---|
| PaddleOCR | 88.7 | 71.9 | 82.1 | 78.7 |
| EasyOCR | 86.5 | 71.5 | 79.1 | 76.6 |

