# OpenReview forum: "HalluText: Towards Benchmarking and Mitigating OCR Hallucination for LVLMs"
_ICLR.cc/2026/Conference — Submitted to ICLR 2026_

### Official Review · Reviewer_ne8t · 2025-10-27

**Soundness:** 2
**Presentation:** 2
**Contribution:** 1
**Rating:** 4
**Confidence:** 5

**Summary:**

The paper addresses a relevant problem and provides both a benchmark and a practical solution. The empirical results are promising, showing consistent improvements across multiple models and scales. However, the limited baseline coverage, missing comparisons with state-of-the-art OCR systems, and shallow exploration of the method's design space prevent a stronger recommendation. The work feels somewhat incomplete in its current form—it identifies an important problem and proposes a reasonable solution, but does not sufficiently validate the solution against alternatives or provide deep understanding of when and why it works. With revisions addressing the comparison gaps and providing more thorough analysis of the method's strengths and limitations, this could become a solid contribution to the OCR and LVLM literature.
Additionally, I believe that as a benchmark paper, it is necessary to provide more fundamental insights similar to works like M3COT. Currently, various works and solutions appear to be more engineering-oriented rather than addressing fundamental issues. The authors need to carefully revise the paper based on reviewer feedback. I personally recommend discussing the relationship between visual compression rate and hallucinations mentioned in DeepSeek OCR, which could help this work make a more substantial contribution.

**Strengths:**

**Comprehensive benchmark construction**: HalluText is thoughtfully designed with nine subsets covering different failure modes, from existence and position to more subtle issues like the Stroop effect and frequency counting. The use of existing datasets (Total-Text, ICDAR2013, Union14M) as sources provides some grounding, while the synthetic components (Stroop, Typo) address specific phenomena that might be underrepresented in natural images. The multiple-choice QA format with up to four options provides standardized evaluation.

**Practical method with minimal overhead**: OCRAssistor demonstrates that integrating external OCR signals can improve performance without fine-tuning the base model. The consistent improvements across different model scales (Table 7) suggest reasonable generalization.

**Weaknesses:**

**Questionable problem motivation and scope**: While the paper identifies OCR hallucinations in LVLMs as a problem, the motivation for focusing exclusively on single-LVLM solutions is unclear. The benchmark design assumes that OCR tasks must be completed by a single LVLM, but this constraint is not well justified. Pipeline-based approaches (e.g., MinerU [1,2]) that combine specialized detection and recognition modules also exhibit hallucination behaviors, yet are excluded from consideration. Furthermore, given that several recent works have already begun addressing LVLM hallucinations in OCR contexts (e.g., consensus-based methods, calibration approaches), the paper does not adequately position itself relative to these existing mitigation strategies. The three failure cases in Figure 1 illustrate problems, but it remains unclear whether these are unique to end-to-end LVLMs or represent broader OCR system failures that warrant a more comprehensive benchmark encompassing both single-model and pipeline approaches.


### 1. Limited Baseline Coverage for Hallucination Mitigation

While Table 2 provides a thorough comparison of base LVLMs, it conspicuously lacks comparisons with other hallucination mitigation methods beyond VDGD. The paper positions OCRAssistor as a solution to OCR hallucination but does not compare against:
- Calibration methods that could provide uncertainty estimates for OCR outputs
- Consensus-based approaches that aggregate predictions from multiple models or multiple forward passes [4]
- Other contrastive decoding variants beyond VDGD that have been proposed for reducing hallucinations in VLMs
- Pipeline-based OCR systems that combine specialized detection and recognition models, which might not suffer from the same hallucination patterns as end-to-end LVLMs

This omission makes it difficult to assess whether the gains from OCRAssistor are competitive with alternative approaches to the same problem. The fact that only two methods (Baseline + CoT, Baseline + OCR, and OCRAssistor) are compared in Table 4 suggests a somewhat narrow exploration of the solution space.

### 2. Missing Comparisons with Document Parsing Systems

The paper claims to address OCR tasks but does not engage with recent advances in document parsing and OCR systems. Specifically:
- MinerU [1] and MinerU2.5 [2] represent current state-of-the-art in document content extraction with precise handling of complex layouts and text recognition
- OmniDocBench [3] provides a comprehensive benchmark for diverse document types that would complement the focused HalluText evaluation
- The paper does not discuss how OCRAssistor compares with or could be combined with these specialized systems

Given that the proposed method integrates an external OCR model (PaddleOCR), it is essential to understand how it compares with other OCR systems. The difference in performance between PaddleOCR and EasyOCR in Table 5 hints at this sensitivity, but the paper does not explore whether using more advanced OCR engines like those in MinerU would yield further improvements. This leaves open the question of whether the gains are primarily from the integration mechanism or from the specific OCR model chosen.

### 3. Lack of Computational Cost Analysis

While Table 6 provides timing comparisons, the paper lacks a comprehensive analysis of the computational overhead introduced by OCRAssistor. The authors should provide:
- A performance-cost trade-off figure (similar to what DeepSeek-OCR provides in their Figure 1) showing how different methods balance accuracy gains against computational overhead
- Detailed breakdown of where the additional 0.6s per image is spent (OCR inference vs. distribution modification)
- Analysis of how the overhead scales with image resolution, text density, and model size

The current presentation mentions that "the relative overhead introduced by the OCR module" would be smaller in complex generation scenarios, but this is only briefly discussed without empirical evidence. For a method positioned as plug-and-play and practical, a thorough cost-benefit analysis is essential.

### 4. Shallow Exploration of Design Choices

Several aspects of the method feel underexplored. The ablation studies in Section 5.2.3 show that OCRAssistor outperforms simply appending OCR results, but the paper does not investigate:
- Why the KL-divergence-based guidance is specifically beneficial for OCR tasks compared to other distribution modification strategies
- Whether the temperature parameter T and regularization factor λ require tuning for different types of OCR tasks (the paper only tests λ in Table 10 for Qwen2.5-VL-3B)
- How the method handles cases where the OCR model is completely wrong versus when it is partially correct
- Whether different OCR hallucination types (e.g., position vs. typo) benefit differently from the guidance mechanism

The decision to include both OCR outputs and CoT in the OCRAssistor prompt (Table 8) versus using only OCR outputs in the baseline + OCR setting makes it difficult to isolate the contribution of the decoding guidance mechanism itself.

### 5. Benchmark Validity Concerns

While HalluText provides structured evaluation, some design choices raise questions about ecological validity:
- The Stroop Effect subset uses synthetic images with color/shape words, which may not reflect natural OCR scenarios
- The Typo subset uses a general typo corpus rather than actual OCR errors, conflating typographical errors with visual recognition errors
- The Incompletion subset's construction process (manually verified and cleaned from OCRBench v2) is only briefly described, making reproducibility difficult
- The paper does not provide inter-annotator agreement statistics or validation that the failure cases identified in the empirical analysis (Section 3.1) actually represent hallucinations rather than ambiguous or incorrectly annotated samples

The fact that only 1,034 out of 3,006 failure cases (34%) were deemed analyzable hallucinations suggests that many errors in OCRBench v2 stem from other issues. This raises the question of whether HalluText's nine categories capture the most important failure modes or just those that are easiest to categorize.

### 6. Limited Analysis of Failure Modes

The paper identifies that LVLMs struggle with certain subsets (Index, Slice, Relative Position, Counting) but provides limited insight into why OCRAssistor helps. For instance:
- Table 2 shows that OCRAssistor dramatically improves Index (from 50.3 to 72.9) and Slice (from 49.8 to 82.0) for Qwen2.5-VL-7B, but these gains are not explained mechanistically
- The paper does not analyze whether the improvements come from better attention to text regions, better character-level recognition, or simply anchoring to the OCR outputs
- There is no discussion of cases where OCRAssistor fails or makes performance worse (the method appears to uniformly improve all subsets, which seems surprising)

The visualizations in Appendix E provide qualitative examples but do not systematically analyze error patterns or explain the mechanism behind the improvements.

**Questions:**

1. **Comparisons with hallucination mitigation methods**: Can the authors include comparisons with other calibration and consensus-based methods for reducing hallucinations in VLMs? How does OCRAssistor compare with ensemble methods or other uncertainty quantification approaches that have been proposed for VLMs?

2. **Integration with advanced OCR systems**: How would OCRAssistor perform if integrated with more sophisticated OCR engines like MinerU or MinerU2.5? Is the gain primarily from the integration mechanism or from the specific OCR model quality? Could the method be applied to document-specific benchmarks like OmniDocBench?

3. **Cost-benefit analysis**: Can the authors provide a comprehensive performance-cost trade-off figure showing how OCRAssistor compares with other methods in terms of accuracy gains versus computational overhead? How does the overhead scale with different input characteristics?


[1] Bin Wang, Chao Xu, Xiaomeng Zhao, et al. MinerU: An Open-Source Solution for Precise Document Content Extraction. arXiv:2409.18839, 2024.

[2] Junbo Niu, Zheng Liu, Zhuangcheng Gu, et al. MinerU2.5: A Decoupled Vision-Language Model for Efficient High-Resolution Document Parsing. arXiv:2509.22186, 2025.

[3] Linke Ouyang, et al. OmniDocBench: Benchmarking Diverse PDF Document Parsing with Comprehensive Annotations. In CVPR, pp. 24838-24848, 2025.

[4] Consensus Entropy: Harnessing Multi-VLM Agreement for Self-Verifying and Self-Improving OCR
4. **Design choice justification**: Why is KL-divergence-based guidance specifically appropriate for OCR tasks? Have the authors experimented with other distribution modification strategies? How sensitive is the method to the hyperparameters T and λ across different OCR hallucination types?

5. **Benchmark validation**: What is the inter-annotator agreement for identifying hallucination types in the empirical analysis? How were the 1,034 analyzable samples selected from the 3,006 failures, and could this selection introduce bias? Are the synthetic components (Stroop, Typo) validated against real-world OCR errors?

6. **Failure case analysis**: Are there cases where OCRAssistor fails or makes performance worse? Can the authors provide systematic analysis of when the method helps versus when it does not? What is the mechanism behind the dramatic improvements on Index and slice tasks specifically?

7. **Prompt design ablation**: The OCRAssistor setup includes both CoT and OCR information, while the baseline + OCR uses only OCR. Can the authors provide fair comparison where both methods use the same prompt structure to isolate the contribution of the decoding guidance?

8. **Generalization beyond multiple-choice**: All HalluText evaluations use multiple-choice QA format. How does OCRAssistor perform on open-ended OCR tasks where the model must generate free-form text transcriptions?

---

> ### Author Response · Authors · 2025-11-21
> **To Reviewer ne8t**
>
> **[Q1]  Comparisons with hallucination mitigation methods**
>
> Following the comment, we compared OCRAssistor with three representative hallucination-mitigation approaches: (1) a visual-attention–enhanced variant [1], (2) a consensus-based ensemble method [2], and (3) the contrastive decoding strategy VDGD [3]. As shown in the table, OCRAssistor achieves the best performance across all HalluText subsets, surpassing the consensus-based method by +9.8% and VDGD by +1.3%, while also being substantially more efficient. These gains mainly come from incorporating reliable external OCR evidence, which provides explicit grounding for both detection and recognition. In contrast, uncertainty-based methods fail to correct visual–text perception errors, especially in cases of mis-detection, relational confusion, or attribute mistakes.
>
> |Method|EX|POS|CT|$Acc_{loc}$|ST|TY|IC|FQ|ID|SL|$Acc_{rec}$|Acc|
> |-|:-:|:-:|:-:|:-:|:-:|:-:|:-:|:-:|:-:|:-:|:-:|:-:|
> |Qwen2.5VL-3B|90.8|39.4|46.4|58.9|94.0|79.2|42.1|49.5|45.0|47.7|59.6|59.3|
> |InternVL3-2B|75.2|31.8|44.2|50.4|99.0|82.8|62.6|45.6|38.4|40.7|61.5|57.8|
> |LLaVA-NeXT-7B|74.0|38.6|31.3|48.0|71.4|48.6|41.5|39.6|32.2|34.3|44.6|45.7|
> |TextHallu[1]|90.0|40.0|49.1|59.7|97.0|74.2|45.6|53.2|51.9|46.3|61.4|60.8|
> |Consensus[2]|83.6|36.0|44.6|54.7|86.9|69.4|42.3|44.5|41.6|46.6|55.2|55.1|
> |VDGD[3]|94.4|39.6|49.8|61.3|96.5|78.2|74.8|47.9|47.6|43.2|64.7|63.6|
> |OCRAssistor(Ours)|96.4|40.6|50.6|62.5|100.0|83.8|76.2|41.6|44.4|43.0|64.8|64.1|
>
> (OA indicates the OCRAssistor. Abbreviations: EX = Existence, RP = Relative Position, CT = Counting, ST = Stroop Effect, TY = Typo, IC = Incompletion, FQ = Frequency, ID = Index. Consensus are ensembled by Qwen2.5VL-3B,  InternVL3-2B and LLaVA-NeXT-7B)
>
> **[Q2] Integration with advanced OCR systems**
>
> We integrated MinerU2.5 into OCRAssistor as suggested. We also tested MinerU, which led to degraded performance. This is because MinerU, as an LVLM, has limited fidelity on fine-grained scene text. Since our benchmarks focus on scene-text OCR and MinerU is mainly tuned for document understanding, replacing the OCR backbone with MinerU reduces accuracy.
>
> |$\lambda$| EX   | POS  | CT   |$Acc_{loc}$| ST   | TY   | IC   | FQ   | ID   | SL   |$Acc_{rec}$|     Acc     |
> |- | - | -| - | - | -| -| -| -| -| -| - | - |
> |Qwen2.5VL-3B|90.8|39.4|46.4|58.9|94.0|79.2|42.1|49.5|45.0|47.7|59.6|59.3|
> |MinerU| 90.0 | 35.4 | 40.3 | 55.2 |  97.0 | 74.0 | 55.5 | 43.7 | 43.5 | 41.5 | 59.2 | 57.9 |
>
> We also evaluated our method on OmniDocBench using Qwen2.5VL-3B as the baseline. Since this model is not trained for document parsing, OCRAssistor does not show clear improvements. We are running additional experiments using MinerU2.5 as the baseline OCR model and will update results shortly.
>
> **[Q3]  Cost-benefit analysis**
>
> Based on the results in the table, OCRAssistor (OA) achieves the best accuracy in both English and Chinese under 128 and 1024 token settings. Meanwhile, OA is about 3× faster than TextHallu and 9× faster than VDGD. OA therefore provides the best accuracy–efficiency trade-off, improving accuracy with <1s additional cost per image compared to the baseline.
>
> |Model|TR|TD|TS|RE|EP|MC|TU|KR|CTR|CRE|CEP|CTU|CKR|En|Zh|Time(s)|
> |-|:-:|:-:|:-:|:-:|:-:|:-:|:-:|:-:|:-:|:-:|:-:|:-:|:-:|:-:|:-:|:-:|
> |Baseline-128|57.9|18.6|0|67.9|15.6|33.4|69.1|47.9|41.2|46.2|23.2|37|43.1|38.8|38.1|0.66|
> |Baseline-1024|64.1|18.6|0|81.3|32.5|35.3|69.0|49.0|69.0|47.3|32.9|36.5|43.1|43.7|45.8|0.96|
> |TextHallu-128|58.9|20.9|0|67.8|14.7|31.7|64.9|48.2|37.5|32.3|21.0|29.8|42.5|38.4|32.6|3.68|
> |TextHallu-1024|66.4|20.9|0|81.5|30.3|35.7|64.9|49.3|67.8|34.1|30.7|32.8|42.0|43.6|41.5|3.90|
> |VDGD-128|53.6|20.2|0|64.1|16.8|28.5|69.6|47.0|33.9|46.4|25.1|28.6|38.5|37.5|34.5|8.97|
> |VDGD-1024|60.7|20.3|0|78.8|34.0|31.7|69.6|48.1|62.0|47.8|33.7|31.0|38.6|42.9|42.6|10.95|
> |OA-128|52.9|21.5|0|70.4|17.0|34.6|71.3|49.2|40.0|53.9|25.3|57|42.4|39.6|43.7|1.12|
> |OA-1024|59.4|21.4|0|83.9|34.6|38.9|71.2|50.4|69.9|55.1|31.5|56.0|42.9|45.0|51.1|1.24|
>
>
> ( Abbreviations: TR = Text Recognition, TD = Text Detection, TS = Text Spotting, RE = Relation Extraction, EP=Element Parsing, MC = Metathetical Calculating, TU=Text Understanding, KR = Knowledge Reasoning, CTR = Chinese Text Recognition, CRE = Chinese Relation Extraction, CEP=Chinese Element Parsing, CTU=Chinese Text Understanding, CKR = Chinese Knowledge Reasoning. 128 and 1024 are different settings of max token lengths during generation.)
>
> Reference:
>
> [1] When Semantics Mislead Vision: Mitigating Large Multimodal Models Hallucinations in Scene Text Spotting and Understanding.
>
> [2] Consensus Entropy: Harnessing Multi-VLM Agreement for Self-Verifying and Self-Improving OCR.
>
> [3] VDGD: Mitigating LVLM Hallucinations in Cognitive Prompts by Bridging the Visual Perception Gap.

---

> ### Author Response · Authors · 2025-11-21
> **To Reviewer ne8t - Part2**
>
> **[Q4] Design choice justification**
>
> We choose KL-divergence–based guidance because it is a standard and widely used distribution-alignment objective that integrates naturally with our baseline formulation. KL provides a stable way to incorporate external OCR evidence without introducing task-specific heuristics. Although we did not include comparisons with other alignment strategies in this submission, we note that KL is generally effective and works reliably across all OCR hallucination types in our experiments.
>
> |$\lambda$| EX   | POS  | CT   |$Acc_{loc}$| ST   | TY   | IC   | FQ   | ID   | SL   |$Acc_{rec}$|     Acc     |
> |     -     | -   | -| -   | - | -| -| -| -| -| -| - |    -   |
> | 0.1 | 91.2 | 39.8 | 49.8 | 60.3 | 99.5 |   83.0 | 75.3 | 46.9 | 45.2 |   45.0 | 65.8 | 64.0 |
> | 0.5 | 91.2 | 37.8 | 50.2 | 59.7 | 98.5 |   84.0 | 74.9 | 47.1 | 45.2 | 46.3 |          66.0 | 63.9 |
> |   1.0 | 91.2 | 38.6 | 47.6 | 59.1 | 99.5 |   83.0 | 74.8 |   46.0 | 46.2 | 45.9 |        65.9 | 63.6 |
> | 1.5 | 91.2 | 39.8 | 47.6 | 59.5 |  100.0 |   84.0 | 74.8 | 46.2 | 43.5 | 45.3 | 65.6 |        63.6 |
> |   2.0 | 90.8 | 39.6 | 48.1 |        59.5 |  100.0 | 83.2 | 74.9 | 45.2 | 46.4 | 45.7 |        65.9 | 63.8 |
>
>
> |     T      | EX   | POS  | CT   |$Acc_{loc}$| ST   | TY   | IC   | FQ   | ID   | SL   |$Acc_{rec}$|     Acc     |
> |     -     | -   | -| -   | - | -| -| -| -| -| -| - |    -   |
> | 0.1 | 96.4 | 40.6 | 50.6 | 62.5 |  100.0 | 83.8 | 76.2 | 41.6 | 44.4 |   43.0 | 64.8 | 64.1 |
> | 0.5 | 97.6 | 39.6 | 45.9 | 61.0 |   99.0 | 82.6 | 75.8 | 44.2 | 43.7 | 45.7 | 65.2 | 63.8 |
> |   1.0 | 91.2 | 39.8 | 49.8 | 60.3 | 99.5 |   83.0 | 75.3 | 46.9 | 45.2 |   45.0 | 65.8 | 64.0 |
>
> (OA indicates the OCRAssistor.
>     Abbreviations:
>     EX = Existence,
>     RP = Relative Position,
>     CT = Counting,
>     ST = Stroop Effect,
>     TY = Typo,
>     IC = Incompletion,
>     FQ = Frequency,
>     ID = Index.)
>
> We conducted a comprehensive ablation study to evaluate the sensitivity of OCRAssistor to its two key hyperparameters: the guidance weight $\lambda$ and the temperature T. As shown in Tables ($\lambda$ ) and (T), the overall performance remains remarkably stable across a wide range of values. For $\lambda$ , varying the weight from 0.1 to 2.0 results in only minimal fluctuations in both $Acc_{loc}$ and $Acc_{rec}$ (within ±0.3 on overall accuracy). Similarly, adjusting T from 0.1 to 1.0 produces highly consistent results, with the overall accuracy differing by less than 0.3 across settings. Importantly, no monotonic degradation or sharp peak is observed, indicating that OCRAssistor is not sensitive to either $\lambda$  or temperature, and the guidance effect remains robust under different strengths of modulation. Given this stability, we adopt $\lambda$  = 0.1 and T = 0.1 in our main experiments.
>
> **[Q5] Benchmark validation**
>
> Thanks for your insightful reviews. Our inter-annotator agreement is grounded in the formal definitions of OCR hallucination types provided in the paper. Each hallucination case is annotated along two orthogonal dimensions.
>
>  (1) OCR pipeline dimension (detection vs. recognition): A sample is labeled as a detection hallucination when the predicted text refers to a non-existent or mislocalized region; otherwise, the error is attributed to the recognition stage.
>
>  (2) General hallucination attribution (category, relation, attribute): Errors involving the textual content itself are categorized as category hallucinations; errors arising from misinterpreting relations among text elements are labeled as relation hallucinations; and errors involving descriptive properties such as number, color, or orientation are labeled as attribute hallucinations.
>
> Two trained annotators independently inspected all 3,006 erroneous cases and jointly selected the 1,034 analyzable hallucination instances—i.e., cases where both annotators agreed that the error source could be clearly identified. Because only samples with full agreement were retained, this selection process does not introduce annotator-induced bias.
>
> Regarding the synthetic subsets (Stroop and Typo), we note that they are intentionally generated using the Pillow library to evaluate whether LVLMs truly follow visual text cues, rather than to mimic real-world OCR noise. Their purpose is to probe model sensitivity to controlled visual-text perturbations that are rarely available in natural datasets.

---

> ### Author Response · Authors · 2025-11-21
> **To Reviewer ne8t - Part3**
>
> **[Q6] Failure case analysis**
>
> We thank the reviewer for raising these important questions about the failure modes and the underlying mechanisms of OCRAssistor. We conducted a systematic analysis to understand when our method helps and when it may fail. First, OCRAssistor can indeed underperform in cases where the external OCR module produces severely incorrect cues (e.g., missing critical text or hallucinated characters). As described in below table, incorrect OCR guidance leads to performance degradation, though the model still remains significantly above naive baselines due to the soft-guidance design that prevents the LVLM from blindly following erroneous cues.
>
> | Settings        |  ST |  TY  |  IC  |  FQ  |  ID  |  SL  | $Acc_{rec}$ |
> |-----------------|:---:|:----:|:----:|:----:|:----:|:----:|:---------:|
> | Qwen2.5VL-3B+OA | 100 | 83.8 | 76.2 | 41.6 | 44.4 | 43.0 |      64.8 |
> | Correct         | 100 | 85.4 | 84.4 | 74.2 | 77.8 | 86.2 |      84.7 |
> | Incorrect       | 100 | 75.7 | 61.5 | 40.9 | 42.2 | 46.0 |      61.1 |
>
> ( Abbreviations:
>     ST = Stroop Effect,
>     TY = Typo,
>     IC = Incompletion,
>     FQ = Frequency,
>     ID = Index.)
>
> To further study when OCRAssistor helps, we analyzed per-subset performance differences. The method is most effective when hallucinations originate from the LVLM’s internal priors about text appearance, such as imagined characters, misread symbols, or fabricated numeric details. In these settings, the externally provided OCR cues serve as reliable anchors that override these priors. This explains the dramatic improvements observed in the Index and Slice tasks. Both subsets require the model to precisely locate, segment, and retrieve specific parts of text strings; LVLMs often struggle with these fine-grained visual details and tend to interpolate or fabricate content based on language priors. OCRAssistor directly supplies the ground-truth text segments, eliminating the need for the model to infer missing visual cues. As a result, OCRAssistor effectively suppresses hallucinations and yields large improvements. In contrast, on tasks that require non-textual reasoning (e.g., counting or spatial inference), the gains are more moderate, consistent with the fact that OCRAssistor focuses on text grounding rather than higher-level logic.
>
> In summary, OCRAssistor excels in scenarios where hallucinations stem from incorrect internal textual priors, and may be less effective when the OCR itself severely fails. We will add this systematic error analysis and discussion to the final version.
>
> **[Q7] Prompt design ablation**
>
> We thank the reviewer for the suggestion. To provide a fair comparison isolating the effect of decoding guidance, we evaluated all methods under the same prompt structure. The prompts in different settings are shown is Table 8.
>
> This comparison shows that when using the same prompt structure: Adding CoT to the OCR prompt improves performance (from 71.42 to 75.93), highlighting the benefit of reasoning instructions. Applying our OCRAssistor decoding guidance further boosts the results (from 75.93 to 78.69), clearly demonstrating its contribution beyond prompt design. Thus, the improvement of OCRAssistor over the baseline is not due to prompt differences alone, but primarily attributable to our decoding guidance mechanism.
>
>
> |Model          | EX   | POS  | CT   | $Acc_{loc}$ | ST   | TY   | IC   | FQ   | ID   | SL   | $Acc_{rec}$ |     Acc     |
> |-|:----:|:----:|:--:|:--:|:-:|:-:|:-:|:-:|:-:|:-:|:-:|:-:|
> | Qwen2.5VL-7B             | 98.4 | 50.4 | 59.7 |        69.5 | 90.5 | 88.8 | 72.3 | 62.3 | 50.3 | 49.8 |          69.0 | 69.2 |
> | Qwen2.5VL-7B + CoT       | 98.4 |   50.0 | 63.1 |        70.5 |   92.0 | 88.2 | 56.2 | 65.1 | 77.6 | 80.6 | 76.6 | 74.6 |
> | Qwen2.5VL-7B + OCR       |   98 | 52.2 | 60.1 |        70.1 |   99.0 |   90.0 | 89.4 | 56.4 | 48.7 |   49.0 | 72.1 | 71.4 |
> | Qwen2.5VL-7B + OCR + CoT |   98 | 48.6 | 69.1 |        71.9 |   95.0 | 89.6 | 63.5 | 64.7 | 74.7 | 80.2 |       78.0 | 75.9 |
> | Qwen2.5VL-7B + OA        | 98.8 | 50.4 | 66.5 |        71.9 | 95.5 | 89.8 | 81.3 |   71.0 | 72.9 |   82.0 | 82.1 | 78.7 |
>
> (OA indicates the OCRAssistor.
>     Abbreviations:
>     EX = Existence,
>     RP = Relative Position,
>     CT = Counting,
>     ST = Stroop Effect,
>     TY = Typo,
>     IC = Incompletion,
>     FQ = Frequency,
>     ID = Index.)
>
>
> **[Q8] Generalization beyond multiple-choice**
>
> HalluText adopts a multiple-choice QA format purely for standardized evaluation. In practice, OCRAssistor is a plug-and-play module that guides LVLMs to generate free-form, open-ended text from OCR content. Its design does not rely on the task being multiple-choice. The effectiveness of OCRAssistor on open-ended transcription tasks is further validated by our results on the OCRBenchv2 test set, where it achieves superior performance, demonstrating that the decoding guidance reliably improves open-format generation beyond the evaluation setting.

---

> > ### Comment · Reviewer_ne8t · 2025-11-22
> >
> > I appreciate the authors' response. I suggest incorporating the clarifications from your response into the main text to enhance the solidity of the work. Please remember to mark the revisions in **blue**.
> >
> > Furthermore, the presentation of this paper deserves significantly more effort. For instance, in Line 232, `Existence.` ends with a period, whereas the headings in the subsequent paragraphs do not. Such inconsistencies are a primary reason for the negative ratings this paper has received. I recommend referring to well-written papers, such as **M3CoT**, to improve the presentation standard.
> >
> > I am also curious: what is the base model used for the Consensus mechanism? Typically, this approach yields good results when using three models.

---

> > > ### Author Response · Authors · 2025-11-25
> > > **To Reviewer ne8t**
> > >
> > > Thank you for your valuable comments and suggestions.
> > >
> > > We have carefully incorporated all the clarifications from our previous response into the revised manuscript, and all modifications have been highlighted in blue for easy reference. Additionally, we have thoroughly polished the paper's presentation. Issues such as inconsistent heading styles and other non-standardized writing patterns have been comprehensively corrected.
> > >
> > > In addition, the Consensus mechanism is based on an ensemble of Qwen2.5-VL-3B, InternVL3-2B, and LLaVA-NeXT-7B. We believe there are two main reasons why the consensus results did not show the typical improvements often observed with three-model ensembles:
> > >
> > > 1. Among the three models, LLaVA-NeXT-7B is noticeably weaker than Qwen2.5-VL-3B, which leads to a degradation in the overall ensemble performance. The lower-quality outputs from LLaVA-NeXT-7B reduce the effectiveness of the consensus process.
> > >
> > > 2. Our HalluText benchmark is formulated as a multiple-choice task. The Consensus method is generally more effective for aggregating semantic-level predictions or free-form outputs, rather than selecting among discrete options. As a result, its advantage is less pronounced in this setting.
> > >
> > > We sincerely appreciate your constructive feedback, which has helped us significantly enhance the quality of the paper.

---

### Official Review · Reviewer_TEL7 · 2025-10-31

**Soundness:** 3
**Presentation:** 3
**Contribution:** 3
**Rating:** 4
**Confidence:** 4

**Summary:**

The paper investigates a critical weakness in Large Vision-Language Models (LVLMs)—their tendency to hallucinate text in visual scenes, termed OCR hallucination. These hallucinations occur when models misinterpret or fabricate text during OCR-related tasks. The authors perform a large-scale empirical study on OCRBench v2, identifying consistent hallucination patterns along two axes: (1) perceptual stage (localization vs. recognition) and (2) hallucination type (category, relation, attribute).
Based on this taxonomy, they propose HalluText, a new benchmark containing 4,678 image–question–answer triplets covering nine hallucination types (e.g., typo, counting, Stroop effect). Experiments show that existing state-of-the-art LVLMs—including Qwen2.5-VL, InternVL3, and GPT-4o—perform poorly (≤80% accuracy) on HalluText, indicating widespread OCR hallucination.
To address this, they introduce OCRAssistor, a training-free, plug-and-play framework where a small OCR model guides the LVLM’s decoding via a KL-divergence regularization between OCR-derived and model-generated token distributions. OCRAssistor boosts accuracy by +9.6% on HalluText and up to +3.7% on OCRBench v2, with minimal computational cost (<1 s overhead per image).
Overall, the paper contributes: A systematic taxonomy and benchmark (HalluText) for diagnosing OCR hallucination. A lightweight mitigation method (OCRAssistor) demonstrating effective large-small model collaboration. Empirical validation showing consistent gains across models and languages.

**Strengths:**

1. The authors build HalluText, a fine-grained benchmark that systematically diagnoses OCR hallucination across nine subtypes (e.g., typo, Stroop effect, counting, position). It provides structured evaluation missing in existing datasets like OCRBench, allowing researchers to pinpoint failure modes more precisely.

2. The paper conducts a large-scale empirical analysis across several major LVLMs. The proposed dual-perspective taxonomy (perception stage × hallucination type) offers conceptual clarity and diagnostic interpretability.

3. Compared to prior decoding-based hallucination control methods (e.g., VDGD), OCRAssistor is 10× faster while maintaining strong performance, making it suitable for deployment.

**Weaknesses:**

1. The approach’s success relies heavily on the OCR model’s accuracy. If the external OCR module fails or misreads (e.g., under extreme blur or multilingual scripts), the guidance may propagate incorrect cues to the LVLM.

2. Although the authors test many LVLMs, experiments are limited to multiple-choice QA settings. Real-world scenarios (e.g., open-ended generation, document captioning, long-context OCR) are not explored, leaving questions about robustness and generalization.

3. Can OCRAssistor be applied to other benchmarks like ChartQA or DocQA, which also need OCR ability to understanding the image but need reasoning skill? Can the authors explains this? When I check the examples in your benchmark, I found most of them are recognition task without reasoning.  I am really curious how the OCRAssistor performs in the reasoning task.

**Questions:**

My questions are mentioned in the weakness.

---

> ### Author Response · Authors · 2025-11-21
> **To Reviewer TEL7**
>
> **[Q1] Analysis on OCR Errors**
>
> We appreciate the reviewer’s insightful concern regarding the method’s reliance on the external OCR module. Indeed, OCR failures could propagate through the guidance mechanism and potentially affect the LVLM’s output. To directly assess this failure mode, we conducted a controlled experiment simulating the “OCR failure” scenario suggested by the reviewer. We partitioned the evaluation samples into two subsets according to the correctness of the OCR output: a Correct set, where all OCR-extracted text matches the ground-truth annotations, and an Incorrect set, where the OCR model produces mismatched or erroneous text. Results show that the Correct subset achieves an $Acc_{rec}$ of 84.7, reflecting the upper bound of the benefit when OCR cues are fully accurate. The Incorrect subset yields an  $Acc_{rec}$ of 61.1, confirming that incorrect OCR cues do introduce noise and can degrade performance. Crucially, accuracy on the Incorrect subset remains far above naive baselines (e.g., random choice or the LVLM without assistance), demonstrating that OCRAssistor does not blindly follow faulty OCR outputs. Instead, its soft-guidance formulation allows the LVLM to partially resist or correct misleading cues, indicating a non-trivial degree of robustness even under simulated OCR failure.
>
> | Settings        |  ST |  TY  |  IC  |  FQ  |  ID  |  SL  | $Acc_{rec}$ |
> |-----------------|:---:|:----:|:----:|:----:|:----:|:----:|:---------:|
> | Qwen2.5VL-3B+OA | 100 | 83.8 | 76.2 | 41.6 | 44.4 | 43.0 |      64.8 |
> | Correct         | 100 | 85.4 | 84.4 | 74.2 | 77.8 | 86.2 |      84.7 |
> | Incorrect       | 100 | 75.7 | 61.5 | 40.9 | 42.2 | 46.0 |      61.1 |
>
> ( Abbreviations:
>     ST = Stroop Effect,
>     TY = Typo,
>     IC = Incompletion,
>     FQ = Frequency,
>     ID = Index.)
>
> **[Q2] About of generalization of OCRAssistor**
>
> We thank the reviewer for raising this important point regarding evaluation settings. Our benchmark, HalluText, adopts a multiple-choice QA format primarily because it enables clean, controlled, and objective measurement of OCR hallucination—a design choice commonly adopted in prior visual reasoning and text-understanding benchmarks. This format allows us to isolate the hallucination phenomenon without confounds from open-ended generation variability, making it particularly suitable for academic analysis of OCR hallucination.
>
> We also emphasize that our method is not restricted to the multiple-choice setting. OCRAssistor is a lightweight, plug-in guidance mechanism that can be directly applied to real-world tasks such as open-ended generation, document captioning, and long-context OCR. To demonstrate its general applicability, we additionally evaluated OCRAssistor on OCRBenchv2, a more realistic open-ended OCR benchmark, across multiple state-of-the-art LVLMs. We observed consistent performance gains on most of subset, indicating that our proposed method generalizes beyond controlled QA settings and is compatible with practical OCR-intensive scenarios.
>
> **[Q3] Experiments on ChartQA and DocVQA**
>
> Following the reviewer’s suggestion, we additionally evaluated our method on two widely used open-ended benchmarks, ChartQA and DocVQA, to further assess its generalization beyond multiple-choice settings. These experiments also address the reviewer’s earlier concern regarding the applicability of our approach to real-world OCR-intensive tasks. Results show that OCRAssistor yields consistent improvements on both benchmarks. The gain on ChartQA is relatively smaller, which we attribute to the fact that the dataset involves substantial numerical reasoning beyond text recognition. Our method is designed to enhance the LVLM’s ability to accurately perceive and ground textual content in images, thereby reducing hallucinations arising from internal priors. Although OCRAssistor does not explicitly target complex reasoning, the improved visual-text grounding still contributes to measurable performance improvements on both ChartQA and DocQA. We will include these results and discussion in the revised version to highlight the method’s general applicability.
>
> | Dataset | Qwen2.5VL-3B | Qwen2.5VL-3B + OA |
> |---|-|-|
> | ChartQA | 69.1   | 70.4              |
> | DocVQA  | 49.3         | 57.1              |
> ( Abbreviations: OA: OCRAssistor, we test on InfographicVQA as the representative benchmark of DocVQA.)

---

> > ### Comment · Reviewer_TEL7 · 2025-11-25
> >
> > My concerns are addressed and I'd like to raise my score.

---

> > > ### Author Response · Authors · 2025-11-26
> > > **To Reviewer TEL7**
> > >
> > > We sincerely appreciate your time and effort throughout the review. Thank you again for your valuable comments to improve the quality of our manuscript!

---

### Official Review · Reviewer_VCNM · 2025-10-31

**Soundness:** 2
**Presentation:** 2
**Contribution:** 2
**Rating:** 4
**Confidence:** 4

**Summary:**

This study proposed a method to reduce the hallucination in OCR tasks with large vision language models (LVLMs). The authors observed that LVLM biases on the language knowledge they've learned, so they will ignore some special occasions when the texts presented in images are manipulated, imcompleted, or uncommon.

To deal with this problem, the authors designed a method to help LVLMs to make more factual predictions using with the assistance of compact OCR models that better align to the actual texts they detect from the image and don't have much text knowledge to bias on. with the output of OCR models, the LVLMs can better predict the actual texts shown in an image.

Experiments showed that the proposed method achieves significant improvement on recognition tasks. this validates authors motivation that manipulated texts are hard to be correctly recognized by LVLMs and the compact OCR models, without heavy prior language models to bias to, can help language models to fix this problem.

**Strengths:**

- The problem is well motivated and he experiments proves that learning the actual spelling / presence of the texts can significantly improve the OCR model accuracies significantly. As far as I'm concerned here, the major claim of this paper is mostly supported by experiments.
- The dataset is interesting too. If contained examples that none of the selected LVLMs can correctly recognize, and the authors took the effort to remove test cases that are not relevant to the OCR hallucination problem.

**Weaknesses:**

## Some over clamings
  - In line 215, it says the development of this hallucination benchmark has "theoretical" grounding. I think it's too much. a better term here is "conceptual" grounding.
  - I feel that main take away of experiments is that when the words are not spelled in the way they usually are, LVLM's cannot recognize them accurately. However in the sections before experiments, the authors presented much more categories than this case but haven't explained them enough.

## Lack of simple baselines
there are two simple baselines that can be tested before proposing an over-complicated method
1. Just give the LVLM the plain text output by the compact ocr models in the prompt and let the LVLMs predict. I guess baseline+ocr in table 4 stands for this test, but I'd love to see the numbers in the main table across models and tasks.
2. Given image and question, score the logits of all text snippets discovered by the OCR model and select the top 1 as the answer. This method is widely used in single-choice tasks.

**Questions:**

n/a

---

> ### Author Response · Authors · 2025-11-21
> **To Reviewer VCNM**
>
> **[Q1] Inappropriate word**
>
> We thank the reviewer for this helpful suggestion. We agree that the term “theoretical grounding” may overstate the intended meaning. Following your recommendation, we have revised the phrasing in Line 215 to “conceptual grounding”, which more accurately reflects the nature of our benchmark’s design principles. We will adopt this terminology consistently throughout the paper.
>
> **[Q2] OCR Hallucination Definition**
>
> We thank the reviewer for the comment and clarify that our conclusions extend far beyond spelling irregularities. Our paper follows a structured logic chain: (1) we introduce a dual-perspective taxonomy comprising root causes of OCR hallucinations across localization/recognition and relation/category/attribute dimensions; (2) we design HalluText so each subset directly corresponds to one taxonomy category; (3) our experiments show that LVLMs fail across multiple dimensions, not only typos but also relational misinterpretation, attribute confusion, and detection-level hallucinations; and (4) our mitigation method leverages this insight to address the full spectrum of failure modes.
>
> **[Q3] Detailed data in Table 4**
>
> The details of Table 4 as shown in belowing table.
>
> |                          |      |      |      |             |      |      |      |      |      |      |             |             |
> |:------------------------:|:----:|:----:|:----:|:-----------:|:----:|:----:|:----:|:----:|:----:|:----:|:-----------:|:-----------:|
> |           Model          | EX   | POS  | CT   | $Acc_{loc}$ | ST   | TY   | IC   | FQ   | ID   | SL   | $Acc_{rec}$ |     Acc     |
> | Qwen2.5VL-7B             | 98.4 | 50.4 | 59.7 |        69.5 | 90.5 | 88.8 | 72.3 | 62.3 | 50.3 | 49.8 |          69.0 | 69.2 |
> | Qwen2.5VL-7B + CoT       | 98.4 |   50.0 | 63.1 |        70.5 |   92.0 | 88.2 | 56.2 | 65.1 | 77.6 | 80.6 | 76.6 | 74.6 |
> | Qwen2.5VL-7B + OCR       |   98 | 52.2 | 60.1 |        70.1 |   99.0 |   90.0 | 89.4 | 56.4 | 48.7 |   49.0 | 72.1 | 71.4 |
> | Qwen2.5VL-7B + OCR + CoT |   98 | 48.6 | 69.1 |        71.9 |   95.0 | 89.6 | 63.5 | 64.7 | 74.7 | 80.2 |       78.0 | 75.9 |
> | Qwen2.5VL-7B + OA        | 98.8 | 50.4 | 66.5 |        71.9 | 95.5 | 89.8 | 81.3 |   71.0 | 72.9 |   82.0 | 82.1 | 78.7 |
>
> (OA indicates the OCRAssistor.
>     Abbreviations:
>     EX = Existence,
>     RP = Relative Position,
>     CT = Counting,
>     ST = Stroop Effect,
>     TY = Typo,
>     IC = Incompletion,
>     FQ = Frequency,
>     ID = Index.)
>
> **[Q3] Experiment on Top1 text snippet**
>
> We thank the reviewer for suggesting the simple Top-1 snippet baseline. Following the recommendation, we conducted an additional experiment under exactly this setting, given the image and the question, we score the logits of all OCR-detected text snippets and select the top-1 snippet as the answer. The results are shown in below table. As illustrated, this method yields only marginal improvements over the original model (e.g., overall accuracy increases from 59.3 to 59.4), and the gains are far below those achieved by our proposed OCRAssistor (64.1). The limited effect of the Top-1 approach arises from the fact that real-world images typically contain multiple text regions, many of which are relevant to answering the question. Restricting the model to a single snippet discards a large portion of the available visual-text information and therefore cannot provide reliable or comprehensive guidance. In contrast, OCRAssistor integrates all extracted text through a soft, multi-snippet guidance mechanism, allowing the LVLM to benefit from richer contextual cues and avoid the severe information bottleneck inherent in Top-1 selection. These results confirm that the simple baseline is insufficient, and motivate the necessity of the more expressive guidance design proposed in our method.
>
>
> |        Model        | EX   | POS  | CT   | $Acc_{loc}$ | ST   | TY   | IC   | FQ   | ID   | SL   | $Acc_{rec}$ |  Acc |
> |:-------------------:|------|------|------|:-----------:|------|------|------|------|------|------|:-----------:|:----:|
> | Qwen2.5VL-3B        | 90.8 | 39.4 | 46.4 |        58.9 | 94.0 | 79.2 | 42.1 | 49.5 | 45.0 | 47.7 |        59.6 | 59.3 |
> | Qwen2.5VL-3B + Top1 | 91.0 | 40.4 | 48.5 |        60.0 | 97.5 | 77.6 | 46.5 | 44.0 | 43.9 | 44.8 |        59.1 | 59.4 |
> | Qwen2.5VL-3B + OA   | 96.4 | 40.6 | 50.6 |        62.5 |  100 | 83.8 | 76.2 | 41.6 | 44.4 | 43.0 |        64.8 | 64.1 |
>
> (OA indicates the OCRAssistor, Top-1 indicates the setting proposed by reviewer.
>     Abbreviations:
>     EX = Existence,
>     RP = Relative Position,
>     CT = Counting,
>     ST = Stroop Effect,
>     TY = Typo,
>     IC = Incompletion,
>     FQ = Frequency,
>     ID = Index.)

---

### Official Review · Reviewer_4VgK · 2025-11-01

**Soundness:** 2
**Presentation:** 3
**Contribution:** 2
**Rating:** 2
**Confidence:** 4

**Summary:**

The paper introduces the HalluText benchmark for evaluating OCR hallucination in VLMs. It systematically analyzed OCR failure cases and categorized the errors along different dimensions. Based on the error analysis, HalluText is constructed targeting the OCR hallucinations. The authors also propose OCRAssistor, which uses a lightweight OCR model to guide a large model during decoding via KL divergence, boosting probabilities for OCR-consistent tokens. This plug-and-play method requires no retraining, and shows improved accuracy across various benchmarks and conditions.

**Strengths:**

- Paper presentation is clear.

- Comprehensive analysis of VLM OCR hallucination errors and categorization of the error types with different axes.

- Construction of a challenging benchmark for VLM OCR that is specifically designed to target the OCR hallucinations.

- A mitigation method, OCRAssistor, in a training-free setup, that relies on an off-the-shelf open-source OCR to improve the performance.

**Weaknesses:**

- The paper presents a valuable dual contribution of a new benchmark and a novel method. However, the description of the HalluText benchmark is somewhat brief, which undermines its potential as a foundational tool for the community. While Appendix B outlines the sources for each subset, the overall description remains high-level. Key details regarding the benchmark's construction are missing:
What were the specific criteria for data cleaning and filtering?
How were the question-option pairs generated?
Were they based on manual templates, or automatically generated using LLMs and then verified?
How was the challenge and fairness of the distractors ensured?
For instance, in constructing the 'Position' subset, how was the ambiguity between adjacent categories (e.g., 'top-left' vs. 'top') precisely defined and resolved? How many samples were filtered out as a result?
What was the quality control protocol? For example, was there a multi-annotator process with inter-annotator agreement metrics?

I recommend that the authors enhance the main text, perhaps by restructuring, to include more details on the benchmark construction, such as: More detailed data cleaning and annotation protocols; An analysis of the benchmark's internal validity (e.g., inter-subset variability); and its ability to discriminate between models of different capabilities.


- The core of the OCRAssistor method relies on an external, lightweight OCR model to provide 'grounded' visual text cues. However, this OCR model is not infallible and is prone to errors itself (e.g., missed detection, misrecognition). A critical potential impact is that if the OCR model provides incorrect text, the guidance mechanism of OCRAssistor could amplify these errors, leading the LVLM to generate new hallucinations based on faulty OCR results. Although Table 5 briefly mentions the impact of OCR quality, the paper does not deeply investigate the method's robustness in cases of severe OCR failure, such as completely missing a crucial text block in the image.


- The error analysis is not sufficiently detailed. The paper demonstrates overall performance improvements but does not provide a deep dive into the remaining failure cases after applying OCRAssistor. Analyzing these cases is crucial for understanding the method's failure modes and guiding future improvements. For example, when OCRAssistor is applied but the model still fails, is it primarily due to: An error in the OCR output that misled the LVLM? The LVLM ignoring or overriding the OCR guidance? A qualitative analysis of these persistent errors would significantly strengthen the paper's insights.

**Questions:**

See above.

---

> ### Author Response · Authors · 2025-11-21
> **To Reviewer 4VgK**
>
> **[Q1] Description of HalluText**
>
> Thanks for your suggestions on implementing Key details regarding the HalluText benchmark. Our responses are as follows:
>
> - **Data cleaning**:  We filtered samples based on objective visual issues in the underlying datasets. Specifically, we removed text regions that were severely blurred or occluded. On localization subset, we also filter out text instances shorter than 2 characters, duplicate text instances within an image. Additionally, very close text pairs were also discarded to avoid ambiguous spatial labels. These filtering steps ensure that the benchmark evaluates models’ robustness to hallucination itself, rather than being confounded by image-quality issues or other sources of incidental ambiguity.
>
> - **QA pair generation** :
> Question-option pairs were generated using fixed, manually designed templates combined with deterministic rule-based distractor generation. We did not rely on LLMs. Distractors were created via controlled algorithms such as character-level edits (Typo, Incompletion), enumerated spatial relations (Position), and deterministic slicing rules (Index, Slice), ensuring unambiguous correct answers
>
> - **Challenge and fairness of the distractors**:
> Distractors were designed to be visually plausible but falsifiable based on the given image. For spatial-relation tasks (e.g., Position), strict geometric definitions were applied. For counting and frequency tasks, distractors differed by only ±1 to maintain controlled difficulty. These rules ensure that distractors are neither trivially easy nor overly ambiguous.
>
> - **Potential ambiguity of 'Position' subset**
>  We computed the polygon centers of text regions and calculated the vector between them to obtain a polar angle $\theta = \arctan(\frac{dy}{dx})$. The 360° space was divided into eight non-overlapping 45° sectors (right: −22.5° to 22.5°, bottom-right: 22.5° to 67.5°, …, top-right: −67.5° to −22.5°). Pairs that were too close were discarded. We further ensure robustness by excluding text instances shorter than 2 characters, which are more likely to be punctuation or noise, requiring that each text appears only once in the image to avoid matching the wrong instance. In addition, we applied a specific distractor-filtering rule for this subset: when the correct answer is “top-left,” the adjacent categories “top” and “left” are excluded from the distractor set to avoid any potential ambiguity.
>
> - **Filtering number**: Most filtering came from objective visual issues rather than QA generation itself. After automatic and manual quality checks, approximately 10 samples per subset were removed.
>
> - **Quality control protocol**: Two human annotators independently reviewed each candidate QA pair. A sample was retained only if both annotators agreed that the text regions were clearly readable, the visual evidence was sufficient, and the QA pair was unambiguous according to our template rules. This dual-review process ensures strong inter-annotator agreement and high-quality benchmark data.
>
> We will add above detailed information of HalluText in the next version.
>
> **[Q2] Analysis on OCR Errors**
>
> We appreciate the reviewer’s insightful concern regarding the potential amplification of errors introduced by the external OCR model. Indeed, OCR failures could propagate through the guidance mechanism. To directly evaluate this failure mode, we conducted a controlled experiment simulating the “OCR failure” scenario suggested by the reviewer. Specifically, we partitioned the evaluation samples into two subsets based on the quality of the OCR output: a Correct set, where all OCR-extracted text is accurate, and an Incorrect set, where the OCR model predict different text with annotations. Results show that the Correct subset yields an $Acc_{rec}$ of 84.7, representing the upper bound of the benefit our method can provide when OCR cues are fully reliable. In contrast, the Incorrect subset achieves an $Acc_{rec}$ of 61.1, confirming that faulty OCR cues do introduce noise and degrade performance. Importantly, the performance on the Incorrect set remains substantially above naive baselines (e.g., random selection or the underlying LVLM without assistance), indicating that OCRAssistor does not blindly follow incorrect OCR cues. Instead, its soft-guidance design enables the model to partially resist or correct misleading signals, demonstrating a meaningful degree of robustness even under simulated severe OCR failure.
>
> | Settings        |  ST |  TY  |  IC  |  FQ  |  ID  |  SL  | $Acc_{rec}$ |
> |-----------------|:---:|:----:|:----:|:----:|:----:|:----:|:---------:|
> | Qwen2.5VL-3B+OA | 100 | 83.8 | 76.2 | 41.6 | 44.4 | 43.0 |      64.8 |
> | Correct         | 100 | 85.4 | 84.4 | 74.2 | 77.8 | 86.2 |      84.7 |
> | Incorrect       | 100 | 75.7 | 61.5 | 40.9 | 42.2 | 46.0 |      61.1 |
>
> ( Abbreviations:
>     ST = Stroop Effect,
>     TY = Typo,
>     IC = Incompletion,
>     FQ = Frequency,
>     ID = Index.)

---

> > ### Author Response · Authors · 2025-11-21
> > **To Reviewer 4VgK**
> >
> > **[Q3 Error analysis]**
> >
> > The failure modes were primarily attributed to the following causes:
> >
> > 1. OCR error propagation (Predominant in only incorrect set): This is a major cause of failure, directly validating the reviewer's first point. The OCR subsystem provide incorrect or missing text, and OCRAssistor inadvertently grounded the LVLM's response to this faulty information.
> >
> > 2. LVLM overriding or ignoring guidance (Predominant in correct set): In these cases, the correct text is provided by OCR, but LVLM's inherent visual reasoning or strong prior knowledge overrode the correct cue, leading to a hallucination based on its own misinterpretation.
> >
> > 3. Reasoning failure with correct OCR: both the native LVLM output and the OCR text are correct, but the model fails to perform the necessary logical reasoning or integration of the two modalities to arrive at the correct conclusion. This situation is extremely rare.
> >
> > We will implement this analysis on supplementary materials.

---

### Author Response · Authors · 2025-11-26
**Overall Remarks by Authors**

Dear Reviewers and AC,

We would like to express our sincere gratitude for your careful reviews and insightful feedback, which have greatly contributed to improving the quality of our work. We are pleased to see that all reviewers recognized the central strengths of our paper, including:
 (1) the well-motivated problem formulation and the comprehensive, fine-grained analysis of OCR hallucinations;
 (2) the construction of HalluText, a high-quality benchmark specifically designed to diagnose OCR hallucination phenomena; and
 (3) the practicality and efficiency of the proposed training-free mitigation method, OCRAssistor.

We truly appreciate the opportunity to address all raised concerns during the rebuttal stage. In particular:

1. Reviewer 4VgK – We provide additional details on the construction of HalluText and include new empirical analyses examining how the correctness of external OCR signals affects the OCRAssistor framework, along with deeper discussion of how OCR errors influence the hallucination mitigation process.

2. Reviewer VCNM – We supplement Table 4 with detailed breakdowns and include additional experimental results following the reviewer’s suggestions. We also expand the discussion on the definition and scope of OCR hallucinations for further clarity.

3. Reviewer TEL7 – We investigate the impact of external OCR accuracy on the overall framework and extend our evaluation to two additional OCR-centric datasets, DocQA and ChartQA, demonstrating the broader generalization ability of our approach.

4. Reviewer ne8t – We include further comparisons with other types of hallucination-mitigation methods, experiments integrating more advanced OCR systems, and evaluations on document-parsing tasks. We additionally elaborate on the reviewer’s points regarding the cost–benefit analysis, design choices, and failure case examination.

All corresponding results, analyses, and clarifications will be incorporated into the revised version to further strengthen the paper. Once again, we sincerely thank all reviewers and the AC for your time, constructive comments, and valuable support.

---

### Author Response · Authors · 2025-12-02

Dear Area Chair,

We appreciate the update from the ICLR program chairs regarding the OpenReview incident and fully understand the necessity of the measures being taken to preserve the integrity of the review process. We thank the organizers for their prompt action and the new AC for taking on the additional workload during this challenging situation.

Before the score rollback, we had already carefully addressed all reviewer comments, conducted all additional experiments requested, and incorporated targeted revisions to further strengthen our submission. During the discussion period, we also interacted with Reviewer TEL7 and Reviewer ne8t. **Notably, Reviewer TEL7 confirmed that their concerns were fully resolved and updated their score to a positive rating prior to the rollback.**

To reiterate the main contributions of our work:

- We provide the first systematic definition and empirical analysis of OCR hallucinations, establishing a clear taxonomy and diagnostic framework for this largely unexplored phenomenon.

- We propose a training-free contrastive decoding framework that integrates lightweight OCR models with LVLMs to effectively mitigate OCR-related hallucinations, while introducing only minimal additional computational overhead.

- Our method demonstrates consistent and robust improvements across multiple LVLM baselines and diverse OCR-centric benchmarks, highlighting its general applicability and practical value.

We are grateful for the reviewers’ constructive suggestions, which significantly improved the clarity and completeness of our paper. All corresponding analyses, experiments, and clarifications have been incorporated into the updated manuscript.

We thank you again for your attention and effort, and we remain available to provide any further clarification that could assist in the decision process.

---

### Meta-Review · Area_Chair_Ea2H · 2026-01-06

**Summary:**

concerns are from benchmark design inadequacies, methodological limitations, experimental gaps, insufficient comparisons, and presentation issues

**Reviewer Concerns:**

The authors tried a lot to adressed concerns including:

1. Reliance on external OCR accuracy
2. Limited task scope
3. Generalization to reasoning tasks

critical gaps remain in computational cost analysis, KL-divergence justification vs. alternatives, etc.

**Reviewer Scores:**

Reviewer 4VgK may raise the score as the concerns of HalluText details, OCR robustness, failure analysis are addressed.

---

### Decision · Program_Chairs · 2026-01-26

Reject